# Micropropagation and Acclimatization of *Gymnocalycium* cv. Fancy (Cactaceae): Developmental Responses to Different Explant Types and Hormone Conditions

**DOI:** 10.3390/plants12233932

**Published:** 2023-11-22

**Authors:** Carles Cortés-Olmos, Vladimir Marín Guerra-Sandoval, Vicente Blanca-Giménez, Adrián Rodríguez-Burruezo

**Affiliations:** Instituto Universitario de Conservación y Mejora de la Agrodiversidad Valenciana (COMAV), Universitat Politècnica de València (UPV), Camino de Vera s/n, 46022 Valencia, Spain; carlescortes3@gmail.com (C.C.-O.); vlaguesa@posgrado.upv.es (V.M.G.-S.); vblanca@csa.upv.es (V.B.-G.)

**Keywords:** Cactus, propagation, explant, organogenesis, rhizogenesis, callus, in vitro, plant growth regulator

## Abstract

The *Gymnocalycium* genus includes numerous highly valued species in the ornamental plant market and their propagation is usually carried out using traditional methods. However, there is a lack of information regarding the efficiency of micropropagation through in vitro tissue culture techniques on these species. So, with the objective of establishing an efficient micropropagation protocol that allows for optimizing the plant obtaining processes, the morphogenic potential of *Gymnocalycium* cv. Fancy has been studied in this work. For this purpose, plants of two different sizes (medium and large) were used as the starting material, from which three types of explants were obtained (apex, central discs and bases). The effect of three plant growth regulators (6-Benzylaminopurine, BAP; Kinetin, KIN; and Thidiazuron, TDZ) at three different concentrations each were studied, and the number of generated shoots, the frequency of appearance of callogenesis and rhizogenesis by explant and condition were assessed. An efficient protocol based on the use of KIN at 4 µM and central discs as a starting explant was developed. Moreover, the obtained sprouts rooted successfully (especially using BAP at 2 µM), and their subsequent acclimatization was very effective. Furthermore, emergence of a new morphotype is presented, that has not previously been described.

## 1. Introduction

The genus *Gymnocalycium* (Cactaceae) encompasses a plethora of cactus species that generally grow in arid and semi-arid environments of South America, especially in Argentina, Bolivia, Brazil, Paraguay and Uruguay [1,2,3,4,5]. Their wide distribution area contributes to their ecological range and genetic diversity, a reason for which *Gymnocalycium* spp. exhibits a diverse array of botanical features and specific ecological adaptations, found in 50 and 80 different species, in addition to a range of varieties and different forms that can be found intra-specifically [6]. *Gymnocalycium* species typically exhibit a low-growing, globular or cylindrical stem morphology. Their stems are often solitary, although some species may form small clusters (Figure 1a). Also, their spines may differ in size, color and arrangement across species (Figure 1b). In fact, some *Gymnocalycium* species bear long, hair-like spines that provide protection against herbivores and harsh sunlight, while others have shorter and stouter spines. The differentiation in spines can often be used as a means of identifying different species. *Gymnocalycium* spp. are renowned for their showy, often brightly colored flowers. These flowers can be solitary or produced in clusters at the apex of the stem (Figure 1c). The flower shapes, sizes and colors vary widely among species, making them particularly appealing to pollinators [1,7,8].

Moreover, this genus garnered substantial recognition and economic importance in the ornamental plant market due to a combination of aesthetic appeal, adaptability, resilience and cultural value [9,10]. Their diverse stem shapes, striking spinations and showy flowers make them desirable choices for both novice plant enthusiasts and experienced cactus hobbyists [10]. Furthermore, they require minimal care, can adapt to different climates and soil types and have the potential for long lifespans, making them interesting for a broad spectrum of gardening and landscaping projects too [9]. In fact, over the past few decades, the demand for ornamental cacti has increased [11].

In this regard, within the global cactus market, it is estimated that around 1.5 million *Gymnocalycium* plants were sold internationally during 2021 [12], with *Gymnocalycium mihanovichii* (Frič and Gürke) Britton and Rose (with more than 1.2 million plants) being the most demanded species. Traditional propagation methods are commonly used in cactus production, including sexual propagation through seeds, asexual propagation through cuttings and clonal propagation through grafting [11,13].These techniques are specially used for cacti produced in mass, aimed at maximizing production under specific conditions and controlled environments.

Nevertheless, within the realm of ornamental plants, *Gymnocalycium* are not just mass-produced products, they are also prized by collectors. In fact, cactus collectors are drawn to special *Gymnocalycium* plants and are willing to invest in acquiring and cultivating these different, particular or unique cacti. The pursuit of rare and distinctive cactus varieties by customers has created economic opportunities for nurseries and growers specializing in ornamental cacti, creating a lucrative niche market. However, it is important to highlight that most of these species of cactus have limited adaptability to mass production conditions. Thus, finding alternative propagation methods, such as in vitro propagation, can favor the inclusion of other valuable cactus species in the market. In this context, *Gymnocalycium* spp. have become important assets to cactus collectors, and refining the methods of mass production can improve the overall ornamental plant market.

For the efficient and rapid production of plants in large quantities, in vitro cultivation can play a fundamental role. Depending on the regenerative capacity of the species, direct organogenesis, indirect organogenesis [14], or axillary bud sprouting [15,16] can be employed, with the latter being the most commonly used regeneration method among cacti [11]. Axillary buds are located in the areoles, which are typical structures of the cactus family. Areolas are meristematic areas that appear as small protuberances from which spines emerge, develop hair-like or woolly structures that protect the meristem, form floral buds and produce new shoots (Figure 1d) [17]. Therefore, propagation from axillary buds involves isolating a portion of tissue containing at least one areola, from which new shoots can be induced by providing the plant with the appropriate exogenous growth regulator [16].

This is particularly valuable for species of horticultural interest, primarily grown for their edible value and/or utility in the industry. Two prominent examples are the prickly pear (*Opuntia* spp.) [16,18,19,20,21,22], known for its edible pads (nopales) and fruit (prickly pears), and the dragon fruit (*Hylocereus* spp.) [23,24], cultivation of which has extensively been researched [16,19,21,22,25,26,27,28,29,30,31,32,33,34]. On the other hand, it is worth noting that in vitro cultivation provides the opportunity to detect and select infrequent phenotypes [35] that would be difficult to identify under high-density planting conditions. Furthermore, the cultivation conditions themselves can lead to somaclonal variation phenomena [35], resulting in the emergence of new morphotypes with recognizable phenotypic alterations. Naturally, the detection of these infrequent variants is of significant commercial interest.

However, there is a lack of information regarding in vitro micropropagation in ornamental cacti. Therefore, despite its advantages, in vitro propagation of ornamental cacti requires the development of efficient and specific protocols, sometimes even at the genotype level [35,36,37,38]. Consequently, it is crucial to determine which tissues and under what cultivation conditions it is possible to induce responses and overcome challenges related to recalcitrance, hyperhydration and oxidation in the culture medium [39,40,41]. The available information on in vitro propagation of *Gymnocalycium* is scarce and limited. Therefore, although relevant information exists for various species [42,43,44,45,46,47,48,49,50], more comprehensive research is needed to develop efficient and reproducible protocols for in vitro propagation. Additionally, these protocols should be feasible to implement and enhance performance in the ornamental plant industry.

Given that *G. mihanovichii* is the most commercially traded species, this study used *Gymnocalycium* cv. Fancy as the starting plant material. This is a commercial hybrid developed by CACTUSLOFT OE (Cullera, Valencia, Spain), obtained through crosses between *Gymnocalycium mihanovichii* and *Gymnocalycium fiedrichi* (*Werdermann*) Pažout, followed by selections to obtain color variants. Due to its genetic structure, the progeny of this cultivar is heterogeneous, resulting in plants with diverse morphology and coloration, which is particularly intriguing when selecting individuals with distinctive characteristics.

In this experiment, medium (M) and large (L) seedlings were used as explant sources and sectioned into different explant types, including apical (A), basal (B) and central discs (CD) that were assessed. Additionally, plant growth regulators (PGRs) 6-Benzilaminopurine (BAP), Kinetin (KIN) and Thidiazuron (TDZ) were also assessed. PGRs were applied at low (BAP2 [2 µM], KIN2 [2 µM] and TDZ1 [1 µM]), medium (BAP4 [4 µM], KIN4 [4 µM] and TDZ2 [2 µM]) and high (BAP8 [8 µM], KIN8 [8 µM] and TDZ4 [4 µM]) concentrations. In this context, the objective of the current study is to establish an efficient in vitro propagation protocol for *Gymnocalycium* cv. Fancy, aiming to (I) increase plant production in facilities dedicated to intensive cultivation and (II) facilitate the detection and propagation of special or atypical plants that could be directed towards the collector’s market. Through this, we aim to assess the effect of growth regulators on the development of *Gymnocalycium* cv. Fancy explants, information that may even be extrapolated to other *Gymnocalycium* species or other cacti whose populations are threatened or in danger of extinction due to human activity.

## 2. Results and Discussion

### 2.1. Callus

A variation in the number of calluses was observed during the assay, reaching in most cases the maximum number of calluses at the end of the induction period (third month) and decreasing from then on (Appendix A). This decrease in the total number of calluses could be partly related to the subculture of the explants to a basal culture medium in absence of PGRs. In fact, morphogenetic responses sometimes do not occur in the presence of PGRs and adventitious shoots only appear upon a transfer onto MS media [13,51]. In the current case, approximately 25% of calluses accounted for in the third month led to defined shoots at the end of the sixth month. The t test verified that the frequency of appearance of calluses was statistically significant only in those conditions in which the TDZ was present. The rest of the PGRs were not able to induce callus formation in a significant proportion (Appendix A). Thus, for callus formation, only the results of TDZ treatments will be analyzed in detail.

From the first month, the apical activation could be detected in all three different TDZ concentrations, but also, bases and central discs were induced during the culture, reaching the highest values at the third month. As apices, bases and central discs presented different average of numbers of areoles (Table 1), and their expected response capacity should be different. Thus, the results could be explained from the perspective of productivity (number of calluses obtained per explant) and efficiency (ratio of activated calluses depending on the number of areoles of each explant).

#### 2.1.1. Productivity of TDZ Treatments Related to Callus Production

Considering the three different TDZ concentrations evaluated, apical explants always offered a better response than bases or central discs when they were cultivated under the same conditions (Figure 2). This indicates that apices were the most productive explants during the study regarding callus production. The response obtained in the apices exposed to TDZ1 stood out since the start of the trial. These apices showed the maximum number of calluses of all the experiments at the third month (Figure 2). From then on, its number of calluses decreased until the end of the assay.

On the other hand, the effect of the initial size of explants was also evaluated. To avoid bias in the results, the response to the different treatments was studied in detail by the type of explant. As central disk explants were only derived from large plants, they were not included in this comparison. The values used for this comparison corresponded to the number of calluses detected at the end of the induction period (third month of the trial), which coincides with the moment of the culture with the highest number of calluses observed.

Our findings showed a higher response of smaller apices than the bigger ones, while no differences between different sizes were observed in basal explants (Figure 3). Both results suggest that younger areoles are probably more responsive than older ones as it has been reported by other researchers [52], a reason for which most protocols for prickly pear micropropagation are usually based on young cladodes [18,19,20,29]. Moreover, larger explants were obtained from plants that germinated earlier and, therefore, also contain older areoles than those present in explants of a smaller initial size. Probably, this also explains why the bases, despite having a greater number of areoles but the oldest ones in the plant, showed lower productivity than the apices and similar to the central discs (Figure 3).

In any case, the relationship between callus production and shoot production may depend on various factors, such as the type of explant, the type and concentration of the PGRs used and other factors related with the in vitro culture conditions [16]. In the present work, the reduction in the number of calluses during the growth period seems to be related to an increase in the number of shoots. However, although this relationship may exist, it does not always occur in other species of cacti [53,54].

#### 2.1.2. Efficiency of TDZ Treatments in Relation to Callus/Areole Production

The central disc efficiency was significantly higher than those observed for apices and bases, with percentages of activated areoles that produced calluses higher than 30% (Table 2). In this regard, more than 47% of the areoles were activatedin the central discs in the presence of TDZ1. By contrast, areolar activation in apices and bases was significatively lower and the averages ranged between 9.66% and 14.58% (Table 2).

Once again, significant differences were found between the efficiency shown by the medium-sized apices and the large-sized apices (Figure 4). In contrast, no differences were detected between the bases of different sizes. This fact reinforces the discussion proposed above, as it seems that younger areoles within the same explant type usually respond better to the treatments than older ones. Taking into account that the apices have the youngest areolas, these results might seem contradictory. However, the ability to transport endogenous hormones and the metabolic capabilities of the cells are tissue-dependent [55,56], so the CD could be more predisposed to areolar activation. Furthermore, the absence of an apical bud in the central discs could be favoring the activation of a greater proportion of areolas, as has been observed in other studies [9,16,57], hence, the greater efficiency in the central discs may not be solely related to the age of the areolas.

### 2.2. Shoots

#### 2.2.1. Productivity Related to Shoot Production

The results showed statistically significant effects of the different types of explants from the second month until the end of in vitro culture period (Table 3). Furthermore, the contribution of the different hormones in the medium was higher during the induction period, while the variation due to the different hormonal concentrations was observed to a greater extent during the last three months of culture. In general, with the exception of the seedling size (insignificant throughout), the principal factors displayed higher significance toward the end of the trial (Table 3).

The study of each factor showed that the presence of BAP and KIN increased the production of shoots during the induction period, especially in central disc explants (Appendix A). Nevertheless, once the induction period ended, a positive response was detected in the explants subjected to the presence of TDZ, leading to the best results at the end of the trial (Appendix A).

Regarding the hormone concentration, it was observed that the explants established in the control groups (in the absence of PGRs) responded, in general terms, with less efficiency than those subjected to cytokinins. The impact of hormone concentration showed that higher concentrations were less effective at producing shoots, while intermediate concentrations generally allowed for higher rates of shoot production (Appendix A).

It is known that responses to exogenous hormones differ based on the species and explant source. Higher hormone concentrations can influence the occurrence of calluses while preventing shoot development, which has been observed in purple pitahaya (*Hylocereus costaricensis*) [34]. Alternatively, higher rates of shoot production in response to high hormone concentrations has been observed in *Mammillaria* [58].

#### 2.2.2. Efficiency Related to Shoot Production

In terms of efficiency, similar results were observed to those obtained for productivity. Once again, BAP and KIN offered the best values at the end of the induction period, and the response of the central discs was higher than that detected for apices and bases (Appendix A). As the cultivation progressed, the increase in shoot production by the explants exposed to TDZ became evident. Thus, at the end of the growing period, TDZ was established as the hormone that generated the most efficient responses in the explants studied. Alternatively, the presence of exogenous PGRs, regardless of their concentration, increased the shoot efficiency compared to the control treatment (Appendix A). However, specific interactions between the hormone type, explant type and hormone concentration might change responses related to the shoot efficiency (Appendix A). For this reason, with the aim of assessing the specific behavior of all possible combinations throughout the trial, the graphical representation of the total production for each combination of factors has also been carried out (Figure 5).

#### 2.2.3. Shoot Production Based on Treatment Factor Combination

Apical explants grown under BAP2, KIN2, KIN4 and KIN8 conditions and the control group showed averages not different from 0, while the rest of the combinations showed significant effects (Table 4). Regarding the total number of shoots, the responses of the explants exposed to the different treatments were highly diverse, especially depending on the type of explant used (Figure 5). But also, the appearance of new shoots during the induction period was considerably different from the response observed during the later developmental period in the absence of regulators.

### 2.3. General Trends Related to the Induction Period

Throughout the first 3 months (induction period), responses related to shoot productivity and efficiency were generally higher for CD explants and significantly higher by the end of the third and fourth months (Appendix A). The central disc explants cultivated under the KIN4 condition produced a total of 24 shoots, followed by the control and BAP4 groups, with 20 shoots each. BAP2_CD, KIN2_CD and TDZ1_CD established values close to these groups (19, 18 and 16 shoots, respectively), but a lower emergence of new shoots was observed under BAP8 and KIN8 presence (11 shoots each). TDZ2 and TDZ4 concentrations resulted in the lowest values, with nine and three shoots, respectively (Figure 5).

Basal explants showed a lower average production than central discs (Appendix A). However, when evaluating specific combinations, it was observed that BAP8_B produced a shoot emergence (22) comparable to the best ones observed in the central disc. The rest of the groups showed levels below 16 shoots (Figure 5).

As previously mentioned, apices from BAP8, TDZ1, TDZ2 and TDZ4 were the only ones that showed some kind of response within these explants. Even so, their results were much lower than those found in central discs and lower in a general terms than the ones observed in bases (Figure 5).

### 2.4. General Trends Related to the Growth Period

During the growing period, in the absence of PGRs, apices showed the lowest values among all studied conditions. Only apical explants cultivated under TDZ displayed high rates of shoot production, particularly TDZ4_A, with a total production of 48 shoots (highest value together with TDZ1_B) (Figure 5).

The apices showed the lowest mean values in the assay when analyzed as a whole, mainly because only those exposed to TDZ responded significantly. KIN, BAP and even the control group either did not provoke a response in the explants (CG_A, KIN2_A and KIN4_A) or gave rise to the worst responses in the assay (BAP2_A, BAP4_A, BAP8_A and KIN8_A). These results evidenced the relevance of the interaction between the hormones and the type of explant (HxTE), so that the presence of TDZ (regardless of its concentration) involved the greatest responses within the overall set of the evaluated apical explants.

The largest monthly increases in the number of shoots for apical explants were observed under TDZ4 and TDZ2 during the sixth month, while shoot formation under TDZ1 was greater during the fifth month (Figure 5). This could indicate that the induction at higher concentrations of TDZ provokes a delay in forming new shoots, both from newly formed or from pre-existing calluses. Similar results were obtained for *Rauvolfia serpentina* (L.) Benth. ex Kurz plants, in which its subculture to an MS media after induction with different TDZ was related with a positive effect in shoot proliferation [59]. 

Regarding the response of the bases, the importance of the HxTE interaction is once again highlighted. In fact, the best results were obtained in the presence of TDZ (more than 40 shoots at each concentration), while the rest of the hormones produced fewer shoots on average (KIN: 24.7 shoots; and BAP: 31 shoots). The control group provided the worst result among the combinations studied, with a total of 19 new shoots at the end of the growing period (Figure 5). Furthermore, a delay in shoot production was also observed as the concentration of TDZ increased, establishing the maximum increases in the number of shoots for TDZ1_B in the 5th month and later for TDZ4_B, during the 6th month. In this regard, it has been reported that the highest shoot proliferation after TDZ induction was also found after the fifth month in other species such as *Rauvolfia serpentina* and *Aegle marmelos* (L.) Correa [59,60]. Probably, the ability of TDZ to activate endogenous cytokinin biosynthesis or altering cytokinin metabolism could be behind these different effects when variable concentrations of this hormone act over the explants [61,62,63,64,65].

In addition, it was also found that the interaction between the three factors (hormones, H; hormone concentrations, HC; and type of explant, TE) played an essential role in the case of the bases cultivated in the presence of BAP8. In fact, the results obtained for BAP8_B (46 shoots) almost doubled those obtained for BAP2_B and BAP4_B (25 and 22 shoots, respectively) (Figure 5); which demonstrates that the effect of the specific combination of the factors was determinant in the response of the explants of the base type.

On the other hand, the relationship between the results obtained from callus emergence after induction with TDZ and its relative production of shoots at a temporal level was also studied. In this case, it was observed that the general behavior of the explants induced at low concentrations differed with respect to the explants subjected to medium or high TDZ concentrations. Thus, in apical and basal explants induced with TDZ1, a reduction close to 60% in the number of calluses was observed from the third to the fifth month (Appendix A). This decrease in the total of counted calluses coincides with a greater increase in the number of shoots, which begins to be remarkable from the fourth month (Figure 5).

In contrast, the induction of apices and bases with TDZ2 and TDZ4 led to a later increase in the number of shoots, especially prominent during the last month of the trial. However, it should be noted that the number of calluses in these cases did not undergo significant variation during the last three months (i.e., a reduction of around 35% in the presence of TDZ2 and 5% for TDZ4) compared to the variation detected for the TDZ1 condition (Appendix A). Thus, the induction with TDZ1 gave rise to the formation of calluses that, during the growing period, underwent a process of organogenesis, causing shoots. However, and given that the variation in the number of calluses was not so remarkable for the TDZ2 and TDZ4 conditions, it appears that the induction at these medium and high concentrations could (i) favor the gradual emission of shoots through the activation of new areolas that had not previously developed callus or, more probably, (ii) suppose an excessive initial hormonal load on the explants, and only with the advance of time and successive subcultures (circumstance that would reduce the endogenous concentration of TDZ in the explants), they were finally able to respond with the issuance of new shoots, as has been observed in other species [59]. These results were comparable to those obtained for *Mammillaria pectinifera* (Ruempler) F.A.C. Weber and *Pelecyphora aselliformis* Ehrenberg calluses, where it could be observed that they began to produce new shoots after a subculture in a medium in the absence of PGRs [66,67].

The behavior of the central discs induced with TDZ did not fit to the responses observed for apices and bases. The variation in the number of calluses is much lower than the increase in the number of shoots observed during the growing period. This suggests that the presence of TDZ (regardless of its concentration) could trigger the formation of new shoots from apparently inactive areolas.

For central disc explants, a delay in areola activation was also observed as the TDZ concentration increased during induction. But in this case, TDZ1_CD and TDZ2_CD gave rise to a global production of 43 shoots, while TDZ4_CD finished with 30. This difference could corroborate the previous hypothesis, in which an excessive concentration of the TDZ during induction could be delaying the emergence of new sprouts. Even so, considering the high effect of the interaction between hormones and the explant type, it might be possible that TDZ4 simply did not activate central disk-type explants as efficiently as at lower concentrations.

Finally, and taking into account that the central discs responded in a fairly homogeneous way in all of the evaluated conditions (including the control group), the influence of the hormones in the medium was probably not having as much relevance as for the apices or bases explants. Probably, just the removal of the apex (involving the loss of the apical dominance) could cause the natural activation of the dormant buds of the explants, something usual when working with cacti [66,67]. And consequently, this resulted in outstanding sprout productivity values even in the absence of PGRs.

### 2.5. Descriptions of Morphotypes

As already mentioned, *Gymnocalycium* cv. Fancy is a commercial hybrid with a large and heterogeneous genetic background. In fact, progenies from sowings are quite heterogeneous, appearing as plants with different morphologies (Figure 6). Even so, most of the Fancy seedlings showed hybrid characteristics with a greater tendency towards the general structure of *G. mihanovichii*. In this sense, seedlings can be easily grouped based on the prominence of the tubercles, the length of the spines and the general coloration of the plant.

The most expected morphotypes were initially classified as Morphotype 1 (M1) and Morphotype 2 (M2). M1 presented the following characteristics: shoots with the recognizable characteristics of the commercial cultivar, that is, with pronounced tubercles which present a triangular section, elongated spines with a greater angle of curvature and darker coloration, even purplish, especially in the distal area of the tubercle (Figure 7a). M2 presented shoots were characterized by presenting less pronounced and more rounded tubercles, shorter and more erect spines and an intense light-green color with a shiny appearance (Figure 7b). Both morphologies could be representative of the variation usually expected in the progenies of this cultivar. Therefore, they were grouped together for the statistical analyses.

After evaluating the development of the in vitro explants, the appearance of shoots with new unexpected morphotypes that did not correspond to the initial description were observed. These morphotypes showed a particular development, and their presence in greenhouse’s seedlings had not been detected previously. The appearance of different morphological shoot types in cacti grown under in vitro conditions have been also reported by other authors [13,68,69,70], and it is relatively frequent. Although the exact mechanism is still unknown, it seems that the epigenetic factors responsible for regulating gene expression could be behind the appearance of these morphological variants [71].

These new unexpected morphotypes showed the following characteristics: Morphotype 3 (M3) had slightly flattened shoots characterized by showing irregular bumps on their epidermis, a total absence of spines and a light-green coloration. Initially, they do not show any woolly structures typical of the areolas, but as they developed, hairy areas appeared, erratically distributed over the tubercles. In addition, a verrucous-looking tissue also emerged between the shoots´ bumps, which sometimes caused new aberrant offsets, and others continued their irregular development, acquiring a callus-like appearance (Figure 7c). Morphotype 4 (M4) had flattened shoots of a dark-green color with a tendency to accumulate a greater amount of pigments in the basal area of the areoles, with conical tubercles and the presence of large and hairy areoles. In some cases, erect, very short and delicate spines emerged from these areoles. It tended to develop new shoots with similar characteristics, so finally, they evolved to particular caespitose structures (Figure 7d).

The number and specific morphotype of the obtained shoots by condition at the end of the trial are compiled in Table 5. In general terms, more than 80% of the shoots showed the expected morphologies: 75% corresponding to the typical morphology of the Fancy cultivar (M1) and 7–8% to the M2. The remaining 20% were included in the other described morphologies: M3 accounted for 12.38% of the total, while M4 appeared at a frequency of 5.70%. On the other hand, it was observed that the appearance of the different morphotypes was not arbitrary, but it was subject to the effects exerted by the factors studied and their respective interactions (Table 6).

#### 2.5.1. Occurrence of Expected Morphotypes

The expected morphotypes appeared in a higher frequency, exceeding 91% in BAP and reaching a total of 100% in the presence of KIN and in the absence of PGRs, while only 51.9% of the total recorded shoots in TDZ showed the expected morphologies (Figure 8). These results suggest that the presence of TDZ in the medium would be favoring the appearance of new morphotypes; while in the presence of BAP or KIN or absence of PGRs, the explants tend to respond by emitting the expected shoots. Similar results were obtained by Giusti et al. [13], who detected that the use TDZ was associated with callus formation and hyperhydricity of new shoots, while BAP (especially at higher concentrations) and intermediate KIN concentrations showed a high multiplication rate and originated new axillary shoots of quality for *Escobaria minima* (Baird) D. Hunt, *Mammillaria pectinifera* and *Pelecyphora aselliformis*.

Furthermore, the production of M1 and M2 shoots was significantly higher from the central disc and basis explants than that observed for apices (Figure 9). Thus, considering all the results obtained, to maximize production in a shorter time, it can be established that an adequate and efficient method to clonally propagate this commercial cultivar should be based on the use of MS media supplemented with KIN4 and, for their major efficiency, central discs as the type of explant.

From a commercial point of view, these results are very important since establishing the conditions to allow for the maximum production of shoots per explant results in an increase in the efficiency of the procedure and, therefore, in the possibility to optimize the resources and increase the benefits. Plants cultured in vitro usually grow faster and produce shoots, even in plants that grow solitary in the wild, as in the case of *Aztekium ritteri* Boedeker. In this case, plantlets can produce new shoots after 11 months of being cultivated in vitro [51]. The results show that micropropagation is not only efficient but also reliable. So if the main objective is to propagate a specific commercial material (such as an individual selection of some outstanding specimen), this protocol allows for the production of more homogeneous plants in only a few months.

On the other hand, this micropropagation protocol could be very useful for other *Gymnocalycium* or related cactus species. The *Gymnocalycium* genus is included in the second appendix of the CITES agreement (Convention on International Trade in Endangered Species of Wild Fauna and Flora; http://www.cites.org; accessed on 28 September 2023), so its species are not listed as threatened or endangered [72,73]. However, there are always specific populations, made up of few individuals, whose survival is threatened by numerous pressures of anthropogenic origin, such as agricultural and urban expansions, the introduction of exotic grasses [74], the use of herbicides and pesticides in intensive agriculture [75,76,77] or their sale on the black market [78,79,80,81], which leads to a reduction in specific and intraspecific plant biodiversity, especially for those taxonomically and genetically problematic groups [74,82]. Taking into account the efficiency of the methods presented and the little influence of KIN4 on the final phenotype of the shoots, plants available to reintroduce into their habitat could be produced, contributing to the conservation of these vulnerable populations while respecting its genetic integrity. Furthermore, our findings show that it is also possible to propagate *Gymnocalycium* efficiently from central discs in the absence of PGRs. Considering that the presence of PGRs may alter the morphological and physiological characters and cause somaclonal variation [83], this micropropagation system would make it possible to obtain individuals that did not alter the genetic background of the repopulated populations.

#### 2.5.2. Occurrence of Unexpected Morphotypes

M3 and M4 shoots mainly developed in the presence of TDZ and stood out with relevant proportions in the apical explants (Figure 8 and Figure 9). These results could be related to the previous ones obtained for callus formation and its subsequent evolution in defined shoots; and more specifically, in explants subjected to TDZ. It seems that the presence of TDZ during the induction period activates the development of calluses (especially in apical explants), and these calluses, through processes of indirect organogenesis, are defined as M3 and M4 shoots, so that the vast majority of shoots with M3 and M4 are probably preceded by a callus phase. The generation of callous tissue prior to the organogenesis process is a factor that may favor somaclonal variation [84], and it could be expected that these new morphotypes appear under this condition. These results are in agreement with those observed by Giusti et al. [13] for *Escobaria minima* (Baird) D. Hunt, *Mammillaria pectinifera* (Ruempler) F.A.C.Weber and *Pelecyphora aselliformis* Ehrenberg. In this assay, TDZ media gave them both a profuse callusing and an important shoot proliferation, but a high incidence of hyperhydricity was observed in the final shoots, probably related with their higher concentration in the media. These hyperhydration events have been described in other cactus species in the presence of TDZ [13] but are also frequently observed in numerous plant species [85].

The production of these shoot types was not observed in the control groups or in the presence of KIN, and their respective appearances in BAP were lower than those observed for TDZ, but represented 4.49% of the M3 shoots and 29.27% of the M4 shoots (Table 5). Morphological alterations were also observed in new shoots of *Melocactus glaucescens* Buining and Brederoo when using BAP as the inductor phytohormone [70], which means that the response in *Gymnocalycium* cv. Fancy is not an isolated case and specific concentrations of BAP can provoke morphological variations in different species.

Given that M3 and M4 are unexpected morphotypes and would hardly arise under traditional sowing conditions, these results suggest that the presence of TDZ in vitro could be acting as an inducer of somaclonal variation, as has been observed in other works [13,86]. Thus, the indirect organogenesis processes that are observed from the calluses could give rise to new morphotypes that would be of commercial interest. This circumstance can be evaluated from two perspectives: (1) the obtained morphotype presents certain characteristics that make it susceptible to being massively propagated and introduced into the wholesale market, or (2) the obtained morphotype is morphologically strange, slow-growing or requires very special growing conditions, characteristics that limit its cultivation in large quantities. In this case, it could be used for the market aimed at cactus collectors. In any case, new varieties of ornamental plants have been developed through spontaneous mutation in vitro [87,88] and TDZ seems to have an important potential for cactus species in this respect.

There is a very important worldwide market focused on cacti for collectors. This market niche is closely linked to obtaining new hybrids, cultivars or forms that show differential characteristics and that would be likely to be collected by cactus lovers. The protocol described in this assay shows that there is the possibility of forcing the appearance of morphotypes other than the original through the induction of explants with TDZ. Thus, this test shows that the culture of *Gymnocalycium* cv. Fancy, mainly in TDZ concentrations between 1 µM and 2 µM, is really effective in obtaining phenotypic variants. This occurrence of somaclonal variation during the tissue culture has also been reported in other cactus species such as *Mammillaria* sp. [89] and *Cereus peruvianus* [90], but not in such remarkable proportions. Thus, a new alternative to obtain a variation in an efficient way is offered, being a useful breeding tool for those cactus producers and growers whose products are aimed at the collector market.

### 2.6. Root Emergence

Root emergence was also observed in some specimens. The development of roots could impact the occurrence and/or development of new shoots in vitro in addition to affecting their subsequent acclimatization capacity [91,92,93]. Therefore, the root emergence of the different treatments was also evaluated.

The rooting response was evaluated during 8 weeks, as no more explants emitted roots after this period of time, and the accumulated percentage of rooted explants is represented in Figure 10. Four groups were defined, whose values showed statistically significant differences between them: (i) conditions with a response of > 75%; (ii) conditions with a percentage between 25 and 37.5%; (iii) conditions with a response varying from 6.25 to 12.5%; and (iv) no response (0%).

Results revealed that the rooting percentage (number of rooted explants/total explants per treatment) was closely related with the type of explant used but also with the PGRs present in the media (Table 7). In fact, apical explants gave rise to the best results regardless of the hormone concentration used, with ratios higher than 75%, except for those ones cultivated in th presence on TDZ. However, no bases showed significant responses and only BAP2, KIN4 and KIN8 came to significatively stimulate the root development in central discs (Figure 10).

Rooting in the apical explants could be expected since the removal of the basal zone of the plant should make the explant respond like a cutting taken in the greenhouse for traditional propagation. So usually, the root development is only a response that fits the natural behavior of cacti and can occur spontaneously in PGR-free media, as it has also been observed in other cactus species [94]. This fact has been contrasted in our experiment, where the control group (in absence of plant growth regulators) has provided similar values to those obtained in the presence of KIN and BAP.

By contrast, the presence of TDZ completely blocked the emission of roots. These results could be explained from the point of view of the synthetic origin of this plant growth regulator, which presents both auxin and cytokinin activity [95]. In fact, similar results had been reported by Rubluo et al. [96] when they used the artificial auxin 2,4-dichlorophenoxy-acetic acid (2,4-D). Its presence in the media offered lower rhyzogenesis and even provoked a total inhibition of rooting at highest concentration evaluated (27.14 µM), when compared to those explants cultivated in the presence of natural or synthetic auxins (IAA, 3-indolyl-acetic acid; NAA, 1-naphthalene-acetic acid and IBA, indole-butyric acid). Perhaps, the presence of TDZ inhibits the natural rooting mechanisms in this material, while the increase in the concentration of KIN and BAP (since they are natural hormones) does not seem to have a negative impact on this aspect.

On the contrary, the rooting in the basal zone explants was not foreseen. These explants were cultivated in an unnatural position to maximize the surface in contact with the culture medium, so it was expected that this placement on the medium would reduce the possibility of rooting, which has been confirmed in this experiment.

The central discs showed higher percentages of rooting also in the presence of BAP2, KIN4 and KIN8 groups, but far from those obtained for the apices (Table 7). Given that the central discs were sectioned into two halves, it could be that the rooting response was being conditioned by the fact that the vascular bundle of the explant was fragmented. On the other hand, apices from the control group showed similar results to apices in the presence of KIN and BAP, while central discs rooting in the control group were significantly lower than those reached by BAP2_CD, KIN4_CD and KIN8_CD combinations (Figure 10).

These results could reinforce the hypothesis that explants with an intact vascular bundle and in contact with the culture medium (as the apices) can develop roots naturally and do not require the presence of PGRs to offer high percentages of rooting. In fact, it has been observed that the dissection of the plants usually alters the proportion of the endogenous hormones favoring a high auxin/cytokinin ratio in the apical zone, which generally promotes the root production in this kind of explants [97,98]. However, an incomplete or damaged vascular bundle would not respond easily and would require the presence of certain concentrations of hormones in the medium to stimulate its rooting.

Thus, the presence of BAP2 in the medium could optimize the procedure and offered the highest rooting percentages in the shortest time (4 weeks). This additional information regarding the influence of BAP2 on rooting is of high interest at a productive level, since it allows the speeding up of the cloning processes. Therefore, it is possible to reduce the necessary time to obtain optimal plants capable of overcoming the acclimatization processes.

### 2.7. Acclimatization

The acclimatization success rate was very remarkable: 96.67% for M1, 100% for M2, 96.67% for M3 and 93.33% for M4. All plants increased their size and showed a healthy appearance. It was also observed that the plants from the expected morphotypes continued their development, resulting in a set of plants representative of the usual variability found within the studied cultivar (Figure 11). However, M3 showed changes in its development.

The acclimatized plants corresponding to M3 began to revert their morphology toward the expected morphotypes. In fact, all the areoles that were activated ex vitro originated from M1 and M2 shoots (Figure 12a) or, in other cases, the apices modified their development and structure giving rise to tissues corresponding to the expected morphotypes (Figure 12b). It seems evident that M3 is linked to the in vitro culture conditions, a fact that could be mainly explained from two points of view: (i) perhaps the presence of PGRs or the high concentrations of other compounds in the media conditions the expected development of some of the new sprouts, as has been observed in other works [99,100,101]; or (ii) alternatively, some shoots suffer a certain degree of vitrification, causing the modification of their natural appearance. In any case, M3 is an artifact generated by the in vitro culture stress, provoked by the presence of phytohormones in the medium or due to problems related to osmotic stress. For this reason, after the acclimatization process, M3 plants showed total instability, reverted their morphology and continued their development with an expected morphology.

On the other hand, M4 shoots kept their particular caespitose development that they showed during the in vitro culture period. In fact, they gave rise to clustered plants with a differential appearance compared to the expected morphotypes (Figure 13). It would be difficult to determine if this variant is the result of a somaclonal variation process, due to a spontaneous mutation, or simply a rare phenotype within the existing variation that *cv*. Fancy presents. But in any case, it has been possible to identify this new morphology and demonstrate its stability under ex vitro culture conditions. This situation shows that the micropropagation of *Gymnocalycium* in vitro can also be tremendously interesting when it comes to identifying particular phenotypes within heterogeneous populations and facilitating their subsequent propagation.

## 3. Materials and Methods

### 3.1. Plant Material and Disinfection

Plant material was obtained by in vitro germination of seeds of *Gymnocalycium* cv. Fancy, kindly donated by Cactusloft OE (Cullera, Valencia, Spain). Seeds were disinfected for 1 min in 70% ethanol (*v*/*v*), followed by 25 min in 15% domestic bleach solution (*v*/*v*; 4% sodium hypochlorite) supplemented with 0.08% Tween-20 (*v*/*v*) and rinsed 3 times in distilled sterilized water under aseptic conditions in a laminar flow cabinet (model AH-100, Telstar, Terrassa, Spain).

### 3.2. In Vitro Establishment and Culture Conditions

Disinfected seeds were sown in Petri dishes using commercial Murashige and Skoog media [102] at half strength (1/2MS, 2.2 g L^−1^) supplemented with sucrose (15 g L^−1^) and agar (7 g L^−1^), adjusting the pH to 5.7 before autoclaving at 120 °C for 20 min [103]. Seedlings were maintained under in vitro conditions and kept in a growth chamber at 26 ± 2 °C, 16 h day length and photosynthetic photon flux of 50 μmol m^−2^ s^−1^ for 8 months. Seedlings were subcultured monthly to a fresh media to promote their development.

### 3.3. Induction and Tissue Culture Conditions

To assess the morphogenic potential of *Gymnocalycium* cv. Fancy, the effect of different concentrations of three cytokinins on shoot induction and production was studied: 6-Benzylaminopurine (BAP), Kinetin (KIN) and Thidiazuron (TDZ). Plant growth regulators (PGRs) were acquired from Duchefa Biochemie Company (RV Haarlem, The Netherlands). For this purpose, 176 plants with diameters ranging from 8 mm to 16 mm were selected from the set of 8-month-old seedlings.

The selected plants were divided into two groups based on their initial size. The first group (M) consisted of medium-sized plants (with diameters ranging from 8 to 12 mm) (Figure 14i), while the second group (B) included large-sized plants (with diameters ranging from 12 to 16 mm) (Figure 14a). From the medium-sized plants, two types of explants were obtained: apical and basal (Figure 14j,l). Regarding the large-sized seedlings, additional explants were obtained by dividing a central disc into two halves (Figure 14d,e). In total, three types of explants were obtained for each large-sized plant: apical stem, basal stem and the two halves of the central disc (Figure 14b,e,g).

Explants were placed for their induction on 1/2MS (2.2 g L^−1^), supplemented with sucrose (15 g L^−1^), agar (7 g L^−1^) and different concentrations of three cytokinins, BAP (2, 4 and 8 µM), KIN (2, 4 and 8 µM) and TDZ (1, 2 and 4 µM), adjusting the pH to 5.7. The position of the explants in the culture medium is shown in Figure 14c,f,h,k,m. A medium in absence of PGRs was also included as a control formulation for each type of explant. The induction period lasted for three months, during which time, the explants were grown in the presence of the corresponding hormones. Subsequently, all explants were subcultured to a 1/2MS without PGRs at pH 5.7 to continue evaluating their development.

### 3.4. Experimental Design

Forty-eight explants were studied per concentration of each hormone, including sixteen of each type of explant. In the case of apices and bases, eight explants from medium and eight explants from large-sized plantlets were included, but only explants from large sized plants (sixteen central disc halves) were evaluated in the case of central discs (Table 8). Six explants per type of explant and size (i.e., medium size apical, large size apical, medium size basal and large size basal explant, as well as two halves from the central disc) were cultured on each Petri dish. So, eight Petri dishes were included per hormone treatment, which makes a total of 80 Petri dishes.

The appearance of calluses and the presence of shoots were evaluated monthly for 6 months. Results were explained in terms of productivity (i.e., the total number of calluses obtained per explant) and efficiency (i.e., ratio of activated callus depending on the number of areoles of each explant only in those explants that responded to the treatments). The morphology of the obtained shoots was also recorded. Rooting was observed in some explants during the first 8 weeks, so the percentage of rooting was studied too.

### 3.5. Statistical Analysis

In order to identify the conditions that generated responses in the evaluated explants, data were subjected to a Student’s *t*-test using the Statgraphics 18 X64 statistical software (Statpoint Technologies, The Plains, VA, USA). No sensitive responses were removed from the general analysis to avoid biases in results. A confidence level of 95% was required.

Additionally, a multivariate ANOVA analysis was carried out to assess the impact of the four main factors evaluated in our study. These factors included three different plant growth regulators (KIN, BAP, and TDZ), three different hormone concentrations for each PGR and three different types of explants (apices, bases and central discs), as well as the initial size of the seedlings (medium and large), along with the possible interactions between them. The software used for performing this ANOVA analysis was Statgraphics X64 (Statpoint Technologies, The Plains, VA, USA).

Absolute data (averages, productivity values and general production) and percentage data (percentages and efficiency values) were previously transformed before conducting the analysis using the following formulas:(1)$$\mathrm{Absolute}\;\mathrm{data}\;\sqrt{Y+\frac{1}{2}}$$
*Y* = Single data
(2)$$\mathrm{Percentage}\;\mathrm{data}\;\;\sqrt[{arcsin}]{\frac{percentage}{100}}$$

In addition, comparisons of means were made using the Student–Newman–Keuls test, with a significance level of 5%.

### 3.6. Acclimatization

One-hundred and twenty of the obtained shoots (30 corresponding to each recorded morphotype) larger than 15 mm were isolated and subcultured in 1/2MS medium supplemented with sucrose (15 g L^−1^) and agar (7 g L^−1^), adjusting the pH to 5.7, to promote their root development and initiate the acclimatization process. These shoots were maintained in vitro for 4 weeks, during which time, all of them developed roots.

Shoots were rinsed with distilled water and then transplanted into trays containing a substrate composed of 50% peat and 50% vermiculite. These trays were placed in mini-greenhouses to control environmental humidity. During the first two weeks of acclimatization, humidity was gradually reduced from the initial 100% to ambient humidity, by gradually opening the mini-greenhouse cover. After two weeks of cultivation, the mini-greenhouse covers were completely removed, and the plants continued growing under the growth chamber conditions. These plants were maintained in cultivation for an additional 8 weeks for further study of their development.

## 4. Conclusions

In the present work, an efficient micropropagation protocol has been developed for *Gymnocalycium*, based on the use of central discs as explants and the presence of kinetin (KIN4) or 6-benzylaminopurine (BAP8) in the culture medium. Furthermore, the obtained shoots successfully rooted in vitro (especially under the BAP2 condition), and their subsequent acclimatization to potted cultivation was highly effective (96.67%). The reliability and reproducibility of the protocol make it suitable for use in intensive production nurseries. Additionally, since it does not alter the morphology of the resulting shoots, this protocol could also be extrapolated to different *Gymnocalycium* species (or other cacti), whose populations are threatened. It could also serve as a tool to contribute to their conservation, both for the purpose of reintroducing specimens into their natural habitat and for maintaining certain genotypes ex situ.

Furthermore, it has been observed that the presence of Thidiazuron can induce callus formation, especially in apical explants, and approximately 25% of these calluses eventually developed shoots with unexpected morphologies, which in some cases remained even after acclimatization without PGRs. These results suggest that TDZ, at appropriate concentrations, could be used as an inducer of new stable variation.

Therefore, the optimization of this in vitro micropropagation protocol for *Gymnocalycium* cv. Fancy can be a valuable tool for both efficiently cloning material and for identifying distinctive individuals within heterogeneous populations. So then, this first approach can be used as a guideline to continue working with variegated and colored *Gymnocalycium* plants, which nowadays have an increased relevance in the niche of the market aimed at cactus hobbyists.

## Figures and Tables

**Figure 1 plants-12-03932-f001:**
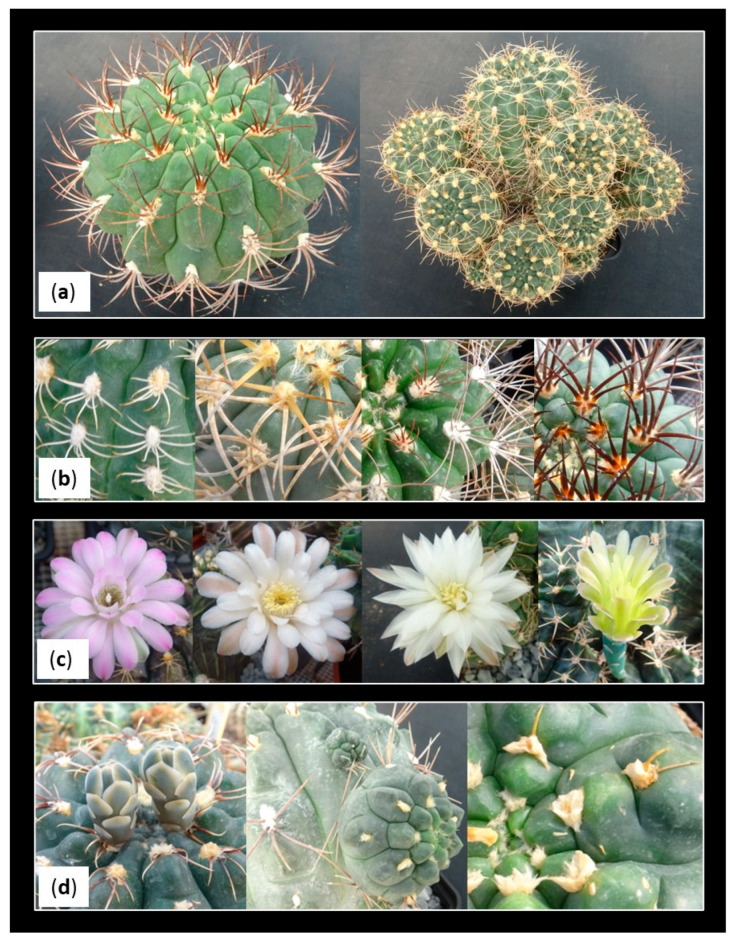
Examples of the variation observed in the genus *Gymnocalycium*. (**a**) Solitary (**left**) and caespitose (**right**) morphologies; (**b**) different kind of spination; (**c**) floral diversity; (**d**) flower buds emerging from the areoles (**left**), new shoots developing from the areoles (**center**) and woolly appearance of some areoles (**right**).

**Figure 2 plants-12-03932-f002:**
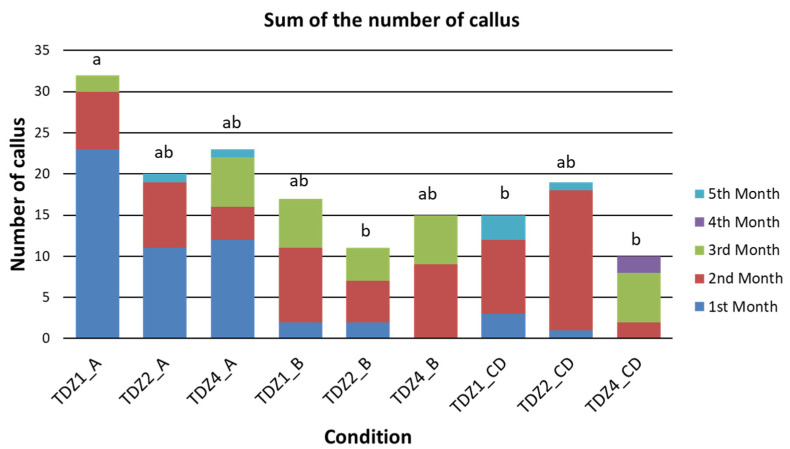
Total sum of callus production by month based on explant type and TDZ concentration. A = apices; B = bases; CD = central discs. Numbers after TDZ (TDZ1, TDZ2 and TDZ4) refer to TDZ concentrations used (1 µM, 2 µM, 4 µM, respectively). Letters (a, b) above the bars represent significant differences at the fifth month based on sample means for *p* = 0.05, according to the Student–Newman–Keuls test.

**Figure 3 plants-12-03932-f003:**
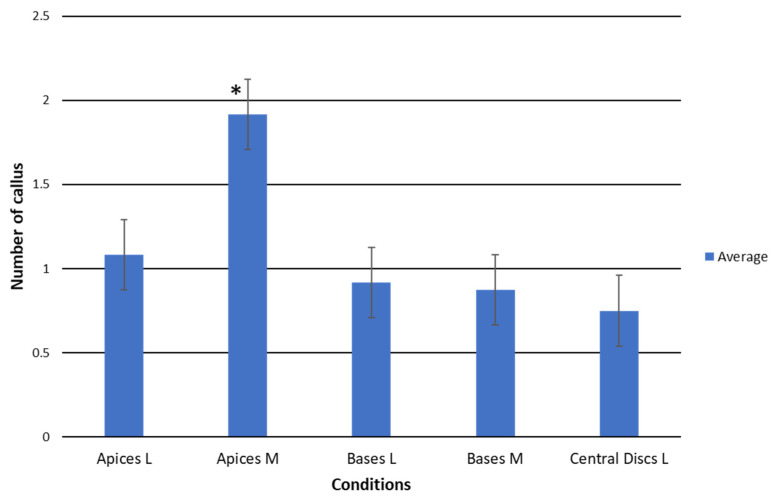
Average number of calluses formed per explant type and initial size of the TDZ treatment at the end of the induction period (third month). Apices L = apical explant from large-size plants; Apices M = apical explant from medium-size plants; Basal L = basal explant from large-size plants; Bases M = basal explant from medium-size plants; Central discs L = central disc explant from large-size plants. The lines in the bars indicate the significant Student–Newman–Keuls difference at *p* < 0.05. ***** = indicates the significatively different condition.

**Figure 4 plants-12-03932-f004:**
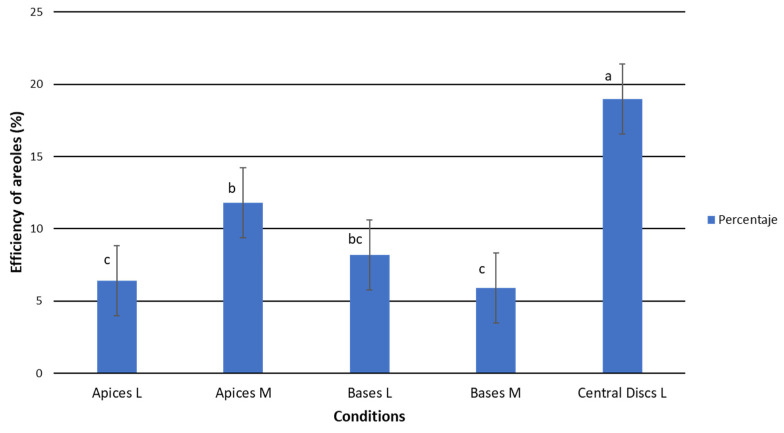
Average efficiency (N° activated areoles/N° total areoles)(×100) by explant type as a function of initial explant size in the TDZ treatments. Apices L = apical explant from large plants; Apices M = apical explant from medium plants; Basal L = basal explant from large plants; Bases M = basal explant from medium-sized plants; Central discs L = central disc explant from large plants. Letters (a, b, c) above the bars represent significant differences at the fifth month for *p* = 0.05, according to Student–Newman–Keuls.

**Figure 5 plants-12-03932-f005:**
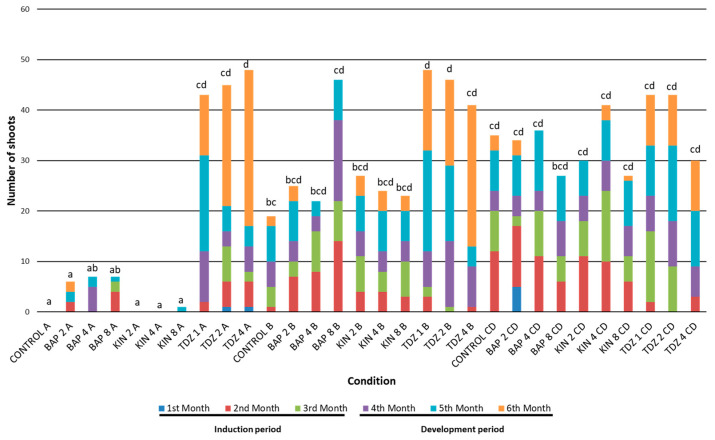
Total shoots produced throughout the experiment based on all factors tested. Conditions: CONTROL (control group), BAP (6-Benzylaminopurine), KIN (Kinetin) and TDZ (Thidiazuron). Numbers following the conditions indicate the hormonal concentration (1, 2, 4 or 8 µM). Capital letters indicate the explants used in each combination: A (apices), B (bases) and CD (central discs). Letters (a, b, c, d) above the bars represent significant differences based on sample means at the sixth month of evaluation, for *p* = 0.05 according to the Student–Newman–Keuls test.

**Figure 6 plants-12-03932-f006:**
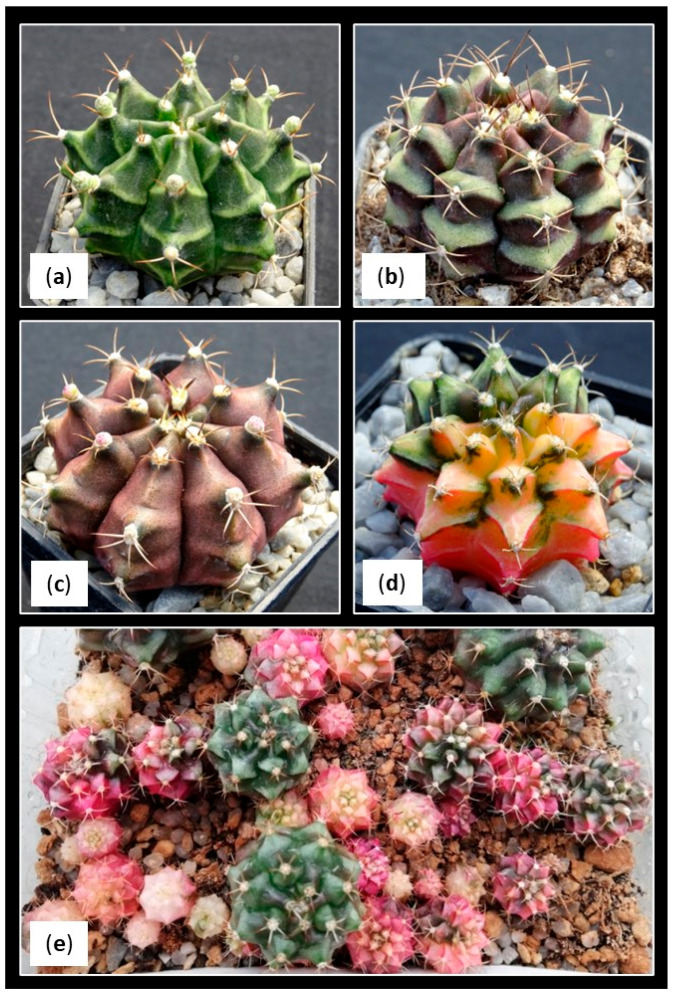
Morphological heterogeneity observed within *Gymnocalycium* cv. Fancy. Four-year-old plants with different colors and shapes are shown (**a**–**d**). One-year-old seedlings are shown (**e**), illustrating segregation in the progeny.

**Figure 7 plants-12-03932-f007:**
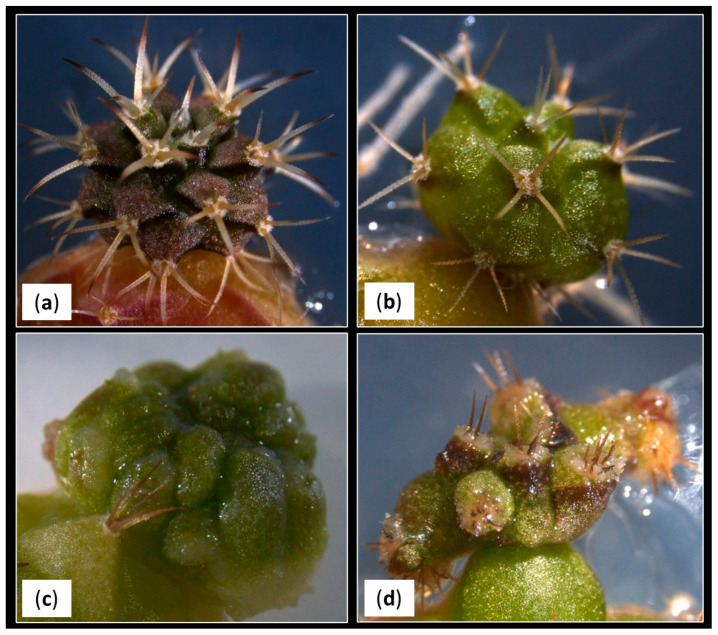
Different morphotypes observed during the trial: (**a**) morphotype 1 (M1); (**b**) morphotype 2 (M2); (**c**) morphotype 3 (M3); (**d**) morphotype 4 (M4).

**Figure 8 plants-12-03932-f008:**
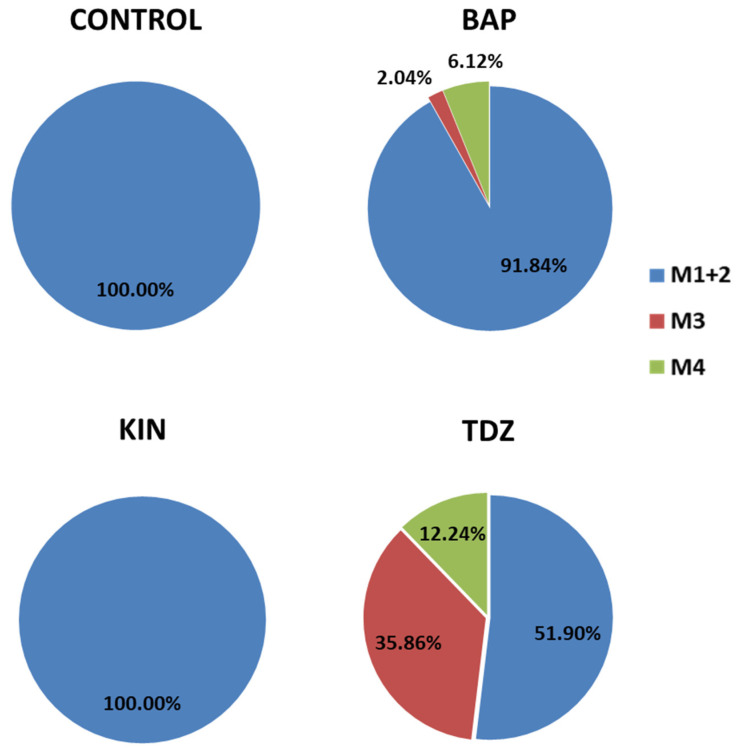
Percent occurrence of each morphotype based on hormone type. BAP = 6-Benzylaminopurine KIN = Kinetin; TDZ= Thidiazuron; M1 + M2 = expected morphotypes (Morphotype 1 plus Morphotype 2); M3 = Morphotype 3 and M4 = Morphotype 4.

**Figure 9 plants-12-03932-f009:**
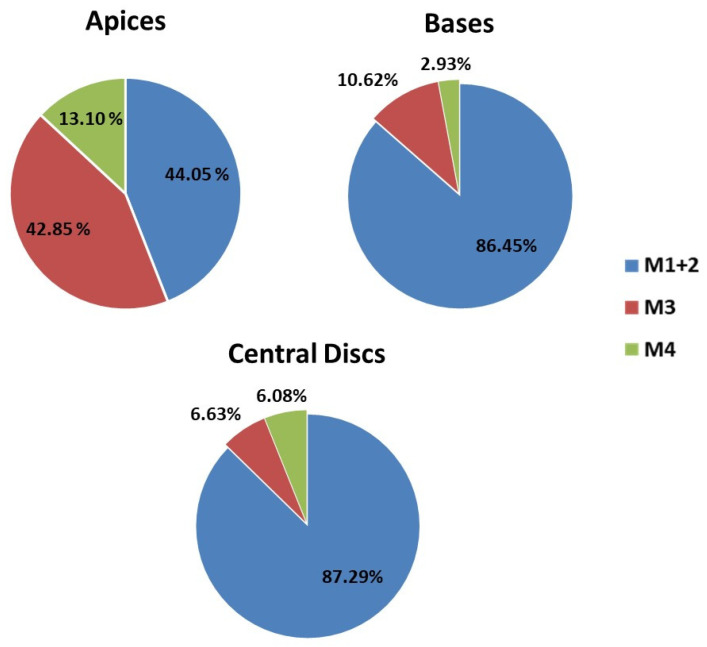
Percent occurrence of each morphotype based on explant type. M1 + M2 = Morphotype 1 plus Morphotype 2; M3 = Morphotype 3 and M4 = Morphotype 4.

**Figure 10 plants-12-03932-f010:**
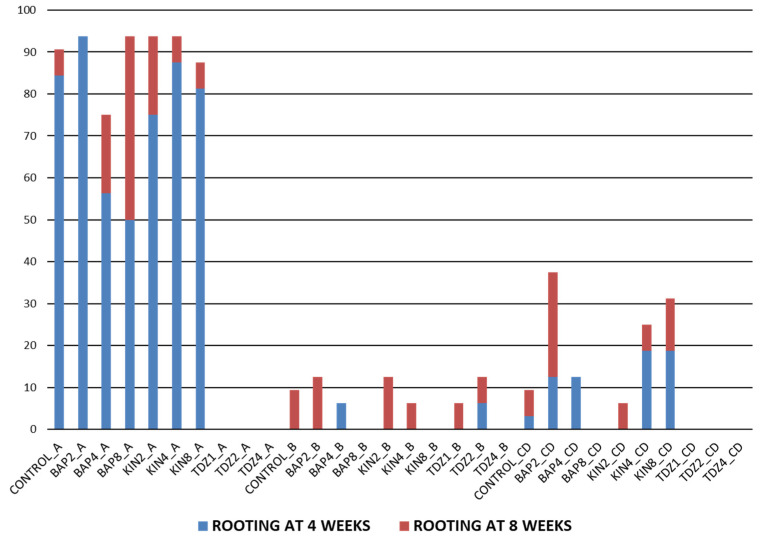
Percentage of rooting explants per combination at 4 and 8 weeks. Conditions: CONTROL (control group), BAP (6-Benzylaminopurine), KIN (Kinetin) and TDZ (Thidiazuron). The numbers following the conditions indicate the hormonal concentration (1, 2, 4 or 8 µM) to which the explants have been subjected. Capital letters indicate the explants used in each combination: A (apices), B (bases) and CD (central discs).

**Figure 11 plants-12-03932-f011:**
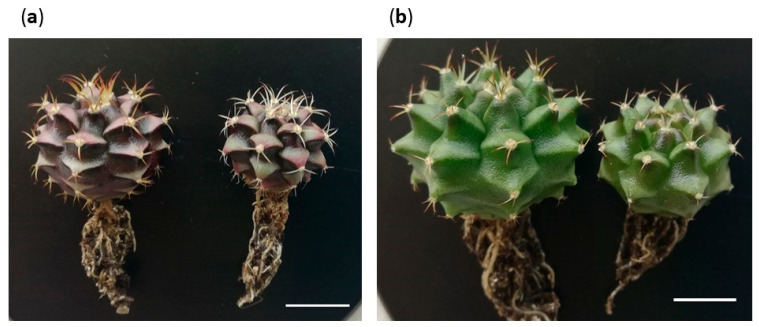
Plants of Morphotypes 1 (**a**) and 2 (**b**) after acclimatization. White bars = 10 mm.

**Figure 12 plants-12-03932-f012:**
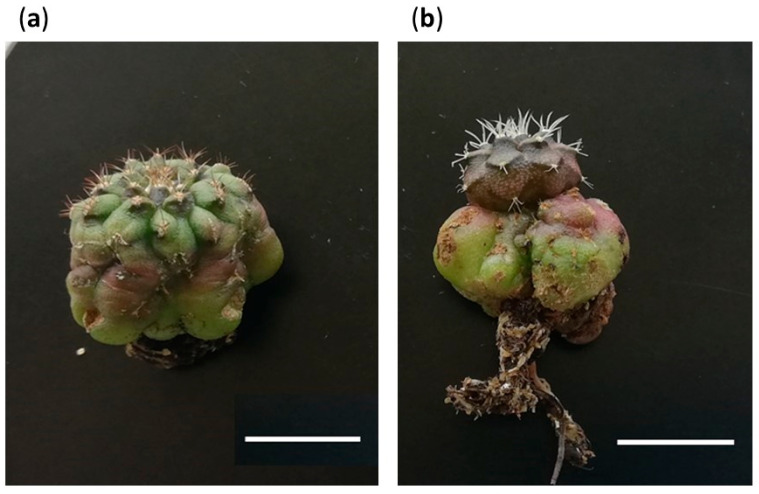
Plants of Morphotype 3 after acclimatization. (**a**) Plants with areoles that were activated ex vitro and originated shoots with morphologies similar to Morphotypes 1 and 2. (**b**) Plants that have modified their development and structure, generating tissues corresponding to the expected morphotypes. Bars = 10 mm.

**Figure 13 plants-12-03932-f013:**
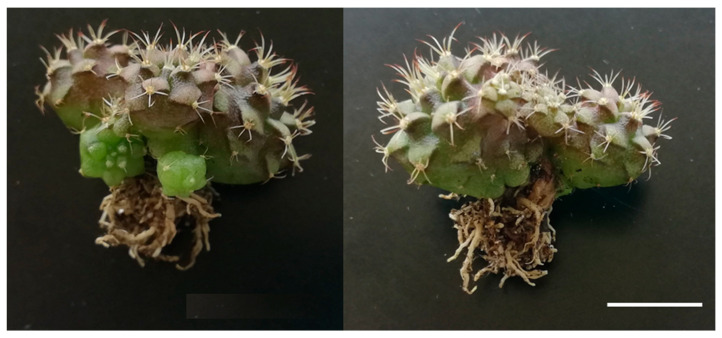
Plants of Morphotype 4 after acclimatization. White bars = 10 mm.

**Figure 14 plants-12-03932-f014:**
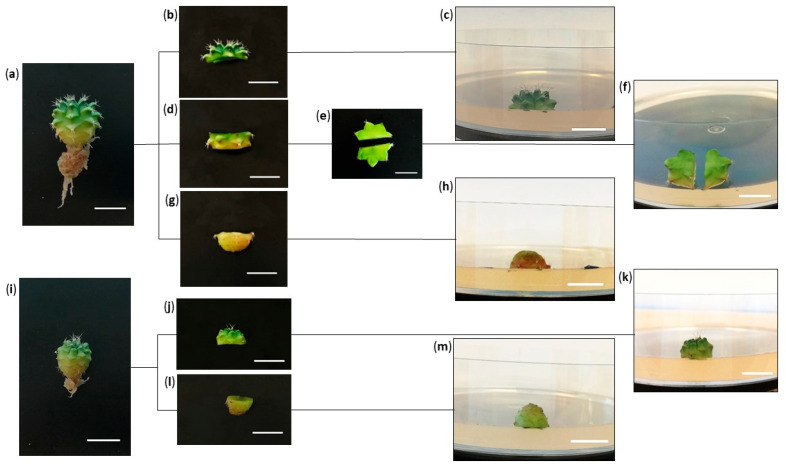
Types of explants of *Gymnocalycium* cv. Fancy. (**a**) Large (L) in vitro germinated plant used as explant source; (**b**) apical explant; (**c**) position of the apical explant in the medium; (**d**) sectioned fragment of the central part of the plant (central disc); (**e**) central disc divided into two halves; (**f**) position of the central discs explants in the medium; (**g**) basal explant; (**h**) position of the basal explants in the medium; (**i**) in vitro germinated plant of medium size (M) used as explant source; (**j**) apical explant; (**k**) position of the apical explant in the medium; (**l**) basal explant; (**m**) position of basal explant in the medium. White bars = 10 mm.

**Table 1 plants-12-03932-t001:** Average number of areolas per explant type of the entire set of evaluated explants.

Explant	Total Areoles Average	Initial Size ^(1)^	Areoles Average ^(2)^
Apices	15.32 ± 0.14	L	15.58 ± 0.20
Apices		M	15.07 ± 0.20
Bases	12.70 ± 0.23	L	12.24 ± 0.32
Bases		M	13.17 ± 0.31
Central Discs	4.12 ± 0.09	L	4.12 ± 0.09

^(1)^ M = medium initial size; L = large initial size. ^(2)^ The values are means ± standard errors.

**Table 2 plants-12-03932-t002:** Percentage of activated areolas as a function of explant type and TDZ concentration.

		Total Number of Explants	Activated Explants	% of Activated Explants	% of Activated Areoles ^(1)^
APICES	TDZ1 ^(2)^	16	13	81.25	14.58 c
	TDZ2	16	11	68.75	10.12 c
	TDZ4	16	12	75.00	11.30 c
BASES	TDZ1	16	11	68.75	13.50 c
	TDZ2	16	8	50.00	9.66 c
	TDZ4	16	9	56.25	12.54 c
CENTRAL DISCS	TDZ1	16	7	43.75	47.55 b
	TDZ2	16	10	62.50	32.98 a
	TDZ4	16	7	43.75	35.48 a

^(1)^ Values followed by the same letter are not statistically different for *p* = 0.05 according to the Student–Newman–Keuls; ^(2)^ TDZ1 = Thidiazuron at 1 µM; TDZ2 = Thidiazuron at 2 µM; TDZ4 = Thidiazuron at 4 µM.

**Table 3 plants-12-03932-t003:** Significance values of the main factors according to each month of evaluation.

Principal Factors	*Gl* ^(1)^	Induction Period			Development Period		
		1st	2nd	3rd	4th	5th	6th
Hormones (H)	2	0.311	0.001 *	0.007 *	0.816	0.102	0.000 *
Type of Explant (TE)	2	0.228	0.000 *	0.000 *****	0.000 *	0.000 *	0.000 *
Hormone Concentration (HC)	3	0.319	0.891	0.461	0.008 *	0.006 *	0.000 *
Seedling Size (SS)	1	0.436	0.238	0.497	0.907	0.317	0.855
Total Cases	576						

^(1)^ Degrees of freedom for the corresponding factor. * Indicates significance at *p* ≤ 0.05 (test—F of Fisher). Induction period includes values obtained during the first three months (1st, 2nd and 3rd) in presence of PGRs. Development period includes values obtained during the last three months (4th, 5th and 6th) in absence of PGRs.

**Table 4 plants-12-03932-t004:** Monthly average number of shoots for each condition.

Condition ^(2)^	Induction Period in Presence of PGRs ^(1)^			Development Period in Absence of PGRs		
	1st Month	2nd Month	3rd Month	4th Month	5th Month	6th Month
Control_A	0.00 ± 0.00	0.00 ± 0.00	0.00 ± 0.00	0.00 ± 0.00	0.00 ± 0.00	0.00 ± 0.00
Control_B	0.00 ± 0.00	0.09 ± 0.05	0.34 ± 0.12	0.64 ± 0.31	1.06 ± 0.25	1.22 ± 0.27
Control_CD	0.00 ± 0.00	0.75 ± 0.20	1.25 ± 0.22	1.49 ± 0.32	2.03 ± 0.29	2.19 ± 0.30
BAP2_A	0.00 ± 0.00	0.13 ± 0.13	0.13 ± 0.13	0.13 ± 0.13	0.19 ± 0.14	0.25 ± 0.19
BAP4_A	0.00 ± 0.00	0.00 ± 0.00	0.00 ± 0.00	0.32± 0.16	0.44 ± 0.22	0.44 ± 0.22
BAP8_A	0.00 ± 0.00	0.25 ± 0.14	0.38 ± 0.18	0.38 ± 0.18	0.44 ± 0.20	0.44 ± 0.20
BAP2_B	0.00 ± 0.00	0.44 ± 0.27	0.63 ± 0.34	0.86 ± 0.35	1.38 ± 0.58	1.56 ± 0.56
BAP4_B	0.00 ± 0.00	0.50 ± 0.22	1.00 ± 0.26	1.18 ± 0.36	1.38 ± 0.33	1.38 ± 0.33
BAP8_B	0.00 ± 0.00	0.88 ± 0.31	1.38 ± 0.41	2.36 ± 0.28	2.88 ± 0.82	2.88 ± 0.82
BAP2_CD	0.31 ± 0.22	1.06 ± 0.32	1.19 ± 0.34	1.44 ± 0.29	1.94 ± 0.35	2.13 ± 0.40
BAP4_CD	0.00 ± 0.00	0.69 ± 0.18	1.25 ± 0.23	1.49 ± 0.30	2.25 ± 0.27	2.25 ± 0.27
BAP8_CD	0.00 ± 0.00	0.38 ± 0.15	0.69 ± 0.27	1.12 ± 0.34	1.69 ± 0.44	1.69 ± 0.44
KIN2_A	0.00 ± 0.00	0.00 ± 0.00	0.00 ± 0.00	0.00 ± 0.00	0.00 ± 0.00	0.00 ± 0.00
KIN4_A	0.00 ± 0.00	0.00 ± 0.00	0.00 ± 0.00	0.00 ± 0.00	0.00 ± 0.00	0.00 ± 0.00
KIN8_A	0.00 ± 0.00	0.00 ± 0.00	0.00 ± 0.00	0.00 ± 0.00	0.06 ± 0.06	0.06 ± 0.06
KIN2_B	0.00 ± 0.00	0.25 ± 0.14	0.69 ± 0.20	1.00 ± 0.35	1.44 ± 0.34	1.69 ± 0.43
KIN4_B	0.00 ± 0.00	0.25 ± 0.14	0.50 ± 0.18	0.74 ± 0.22	1.25 ± 0.19	1.50 ± 0.26
KIN8_B	0.00 ± 0.00	0.19 ± 0.10	0.63 ± 0.24	0.86 ± 0.28	1.25 ± 0.25	1.44 ± 0.27
KIN2_CD	0.00 ± 0.00	0.69 ± 0.25	1.13 ± 0.27	1.44 ± 0.29	1.88 ± 0.29	1.88 ± 0.29
KIN4_CD	0.00 ± 0.00	0.63 ± 0.18	1.50 ± 0.27	1.87 ± 0.25	2.38 ± 0.18	2.56 ± 0.26
KIN8_CD	0.00 ± 0.00	0.40 ± 0.16	0.73 ± 0.21	1.06 ± 0.24	1.73 ± 0.27	1.80 ± 0.30
TDZ1_A	0.00 ± 0.00	0.13 ± 0.13	0.13 ± 0.13	0.74 ± 0.23	1.94 ± 0.35	2.69 ± 0.33
TDZ2_A	0.06 ± 0.06	0.38 ± 0.26	0.81 ± 0.39	1.01 ± 0.38	1.31 ± 0.48	2.81 ± 0.54
TDZ4_A	0.06 ± 0.06	0.38 ± 0.26	0.50 ± 0.26	0.80 ± 0.41	1.06 ± 0.35	3.00 ± 0.43
TDZ1_B	0.00 ± 0.00	0.19 ± 0.14	0.31 ± 0.20	0.75 ± 0.34	2.00 ± 0.45	3.00 ± 0.50
TDZ2_B	0.00 ± 0.00	0.00 ± 0.00	0.06 ± 0.06	0.86 ± 0.34	1.81 ± 0.31	2.88 ± 0.36
TDZ4_B	0.00 ± 0.00	0.06 ± 0.06	0.06 ± 0.06	0.57 ± 0.27	0.81 ± 0.26	2.56 ± 0.62
TDZ1_CD	0.00 ± 0.00	0.13 ± 0.09	1.00 ± 0.32	1.44 ± 0.34	2.06 ± 0.44	2.69 ± 0.39
TDZ2_CD	0.00 ± 0.00	0.00 ± 0.00	0.56 ± 0.22	1.12 ± 0.28	2.06 ± 0.35	2.69 ± 0.41
TDZ4_CD	0.00 ± 0.00	0.19 ± 0.19	0.19 ± 0.19	0.57 ± 0.25	1.25 ± 0.32	1.88 ± 0.29

^(1)^ PGRs (plant growth regulators). Values are means ± standard errors. ^(2)^ Conditions: CONTROL (control group), BAP (6-Benzylaminopurine), KIN (Kinetin) and TDZ (Thidiazuron). Numbers following the conditions indicate the hormonal concentration (1, 2, 4 or 8 µM). Capital letters indicate the explants used in each combination: A (apices), B (bases) and CD (central discs)

**Table 5 plants-12-03932-t005:** Number of shoots for each morphotype detected in each test condition after sixth month.

Condition ^(1)^	N° of Explants	M1	M2	M3	M4
Control_A	48	0	0	0	0
Control_B	48	45	0	0	0
Control_CD	48	84	0	0	0
BAP2_A	16	1	0	0	2
BAP4_A	16	4	0	0	0
BAP8_A	16	2	0	3	0
BAP2_B	16	13	9	0	0
BAP4_B	16	20	1	0	0
BAP8_B	16	31	14	0	1
BAP2_CD	16	23	1	1	7
BAP4_CD	16	34	2	0	0
BAP8_CD	16	17	8	0	2
KIN2_A	16	0	0	0	0
KIN4_A	16	0	0	0	0
KIN8_A	16	1	0	0	0
KIN2_B	16	20	3	0	0
KIN4_B	16	20	0	0	0
KIN8_B	16	10	9	0	0
KIN2_CD	16	30	0	0	0
KIN4_CD	16	36	2	0	0
KIN8_CD	16	22	4	0	0
TDZ1_A	16	7	0	18	8
TDZ2_A	16	11	0	9	1
TDZ4_A	16	11	0	6	0
TDZ1_B	16	20	0	8	4
TDZ2_B	16	18	0	7	3
TDZ4_B	16	3	0	14	0
TDZ1_CD	16	25	1	2	5
TDZ2_CD	16	16	0	16	4
TDZ4_CD	16	11	0	5	4
Total	576	535	54	89	41
Total %		74.41	7.51	12.38	5.7

^(1)^ Conditions: CONTROL (control group), BAP (6-Benzylaminopurine), KIN (Kinetin) and TDZ (Thidiazuron). Numbers following the conditions indicate the hormonal concentration (1, 2, 4 or 8 µM). Capital letters indicate the explants used in each combination: A (apices), B (bases) and CD (central discs). “Total” indicates the sum of number of shoots of each of the observed morphotypes. “Total %” indicates the percentage of shoots corresponding to each of the observed morphotypes.

**Table 6 plants-12-03932-t006:** Significance values of the main factors according to morphotypes.

		Expected Morphotypes (1 + 2)		Morphotype 3		Morphotype 4	
	Cases	*p*-Value ^(1)^	Mean ^(2)^	*p*-Value	Mean	*p*-Value	Mean
**H ^(3)^**		0.01		0.00		0.00	
BAP	192		1.20 ± 0.09 b		0.02 ± 0.03 a		0.06 ± 0.03 a
KIN	192		1.04 ± 0.09 ab		0.00 ± 0.03 a		0.00 ± 0.03 a
TDZ	192		0.82 ± 0.09 a		0.44 ± 0.03 b		0.15 ± 0.03 b
**TE**		0.00		0.45		0.14	
Apices	192		0.19 ± 0.09 a		0.19 ± 0.03 a		0.06 ± 0.03 a
Bases	192		1.21 ± 0.09 b		0.15 ± 0.03 a		0.04 ± 0.03 a
Central Discs	192		1.66 ± 0.09 c		0.12 ± 0.03 a		0.11 ± 0.03 a
**HC**		0.06		0.00		0.00	
Control	144		0.89 ± 0.11 a		0.00 ± 0.04 a		0.00 ± 0.03 a
Low	144		1.06 ± 0.11 a		0.20 ± 0.04 b		0.18 ± 0.03 b
Medium	144		1.14 ± 0.11 a		0.22 ± 0.04 b		0.06 ± 0.03 a
High	144		0.99 ± 0.11 a		0.19 ± 0.04 b		0.05 ± 0.03 a
H × TE		0.00		0.73		0.70	
H × HC		0.12		0.00		0.05	
TE × HC		0.12		0.01		0.58	
H × TE × HC		0.01		0.00		0.48	
Total cases	576						
Total mean			1.021 ± 0.061		0.155 ± 0.023		0.071 ± 0.016

^(1)^ Significant values at *p* ≤ 0.05 (test—F of Fisher). ^(2)^ Values are means ± standard errors. In addition, values followed by the same letter are not statistically different for *p* = 0.05 according to the Student–Newman–Keuls test; ^(3)^ H = hormones; TE = type of explant; HC = hormone concentration; ^2^ BAP = 6-Benzylaminopurine; KIN = Kinetin; TDZ= Thidiazuron.

**Table 7 plants-12-03932-t007:** Percentage of rooting appearance in explants after 4 and 8 weeks.

PGR ^(1)^	Concentration	Type of Explant	Rooting after 4 Weeks ^(2)^	Rooting after 8 Weeks
CONTROL	0	A	84.38 ab	90.63 a
	0	CD	-	-
	0	B	-	-
BAP	2	A	93.75 a	93.75 a
	2	CD	12.50 d	37.50 b
	2	B	-	-
	4	A	56.25 ab	75.00 a
	4	CD	-	-
	4	B	-	-
	8	A	50.00 bc	93.75 a
	8	CD	-	-
	8	B	-	-
KIN	2	A	75.00 ab	93.75 a
	2	CD	-	-
	2	B	-	-
	4	A	87.50 ab	93.75 a
	4	CD	18.75 cd	25.00 b
	4	B	-	-
	8	A	81.25 ab	87.50 a
	8	CD	18.75 cd	31.25 b
	8	B	-	-
TDZ	1	A	-	-
	1	CD	-	-
	1	B	-	-
	2	A	-	-
	2	CD	-	-
	2	B	-	-
	4	A	-	-
	4	CD	-	-
	4	B	-	-

^(1)^ PGRs =Plant growth regulators, BAP = 6-Benzylaminopurine at 2 µM (2), 4 µM (4), 8 µM (8); KIN = Kinetin at 2 µM (2), 4 µM (4), 8 µM (8); TDZ = Thidiazuron at 1 µM (1), 2 µM (2), 4 µM (4); A = apical explant; B = basal explant and CD = central disc explant. Sign (-) indicates that means are not significantly different from zero, according to Student’s *t*-test, and therefore, they are not included in the table. ^(2)^ Values followed by the same letter are not statistically different for *p* = 0.05 according to Student–Newman–Keuls.

**Table 8 plants-12-03932-t008:** Experimental design of the trial.

PGRs ^(1)^	Concentration (µM)	Type of Explant	Initial Size ^(2)^	N° of Explants
BAP	2	Apex	M	8
		Apex	L	8
		Base	M	8
		Base	L	8
		Central Disc	L	16
BAP	4	Apex	M	8
		Apex	L	8
		Base	M	8
		Base	L	8
		Central Disc	L	16
BAP	8	Apex	M	8
		Apex	L	8
		Base	M	8
		Base	L	8
		Central Disc	L	16
KIN	2	Apex	M	8
		Apex	L	8
		Base	M	8
		Base	L	8
		Central Disc	L	16
KIN	4	Apex	M	8
		Apex	L	8
		Base	M	8
		Base	L	8
		Central Disc	L	16
KIN	8	Apex	M	8
		Apex	L	8
		Base	M	8
		Base	L	8
		Central Disc	L	16
TDZ	1	Apex	M	8
		Apex	L	8
		Base	M	8
		Base	L	8
		Central Disc	L	16
TDZ	2	Apex	M	8
		Apex	L	8
		Base	M	8
		Base	L	8
		Central Disc	L	16
TDZ	4	Apex	M	8
		Apex	L	8
		Base	M	8
		Base	L	8
		Central Disc	L	16
CONTROL		Apex	M	16
		Apex	L	16
		Base	M	16
		Base	L	16
		Central Disc	L	32
Total				528

^(1)^ PGRs (plant growth regulators); BAP = 6-Benzylaminopurine; KIN = Kinetin; TDZ = Thidiazuron; ^(2)^ M = medium initial size; L = large initial size.

## Data Availability

Data are contained within the article.

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
