# Peer review of "Micropropagation and Acclimatization of Gymnocalycium cv. Fancy (Cactaceae): Developmental Responses to Different Explant Types and Hormone Conditions"

_plants, 2023, doi:10.3390/plants12233932_

Round 1
Reviewer 1 Report
Comments and Suggestions for Authors
Comments are provided in the attached file.

Many parts of the manuscript are grammatically awkward. Suggestions for improvement are provided in the reviewer comments document submitted.
Author Response
Dear Reviewer,
First of all, many thanks for your contribution to improve our manuscript. In the attached pdf file we include a cover letter addressing all the requested points from you and the second reviewer as well. So that you can have a wider perspective of the revision process.
Nevertheless, below we include exclusively the changes and explanations in response to your comments and suggestions.
Response to REVIEWER 1´s comments
#1. The title was shortened from “Study of the response to in vitro micropropagation of Gymnocalycium cv. Fancy (Cactaceae), a reference cultivar of commercial interest” to “Micropropagation and acclimatization of Gymnocalycium cv. Fancy (Cactaceae): Developmental responses to different explant types and hormone conditions”.
#2. The abstract was written as a single paragraph, considering all the reviewer's suggestions.
#3. Due to the length of the paper, we think that to merge results and discussion in the same section shows our findings in a more dynamic fashion for future readers. We have used this procedure in other similar papers published in MDPI. Therefore, If the reviewer does not mind, we would like to keep this structure to streamline the reading of the paper. Furthermore, we have added more information and new bibliographical references to enrich the discussion in the most salient points, as requested by you and the second reviewer (Reviewer 1, points #27, #34, #74, #77, #80, #81; Reviewer 2, point #6).
#4. Treatment abbreviations have been included in the text following the reviewer's instructions. In addition, it has been unified with the term "large" (instead “big”) when identifying explants that ranged from 12-16 mm, both in text and in figures/tables. Also, the acronym “PGRs” was explained and included in its first use.
#5. All decimals are indicated using “.” (dot, English format), instead “,” (comma, Spanish format).
#6. The entire text has been revised to prevent paragraphs from being shorter than three sentences. Only in very few exceptions, paragraphs with a length of less than three sentences been maintained, either due to their greater length or because joining them to other paragraphs can cause confusion.
#7. Numerical values were only used when necessary. Changes requested by the reviewer have been carried out.
#8. Change made
#9. The clarification regarding the use of "Gymnocalycium genus" is appreciated. This grammatical construction has been replaced by the use of “Gymnocalycium spp.” in all relevant cases.
#10. Change made.
#11. Change made.
#12. In order to clarify the meaning of the sentence, it has been modified from “These techniques are used in intensive crops, aimed at maximizing production under specific conditions and controlled environments.” to “These techniques are specially used for cactus produced in mass, aimed at maximizing production under specific conditions and controlled environments.”
#13. Change made.
#14. Change made.
#15. Change made.
#16. Both sentences have been rewritten based on the reviewer's comments.
#17. Change made.
#18. The presence of the "," (which was a typo) has been removed. Replaced "ssp." for "spp." It was considered appropriate to modify the sentence according to the comments of the reviewer.
#19. Change made.
#20. Space inserted.
#21. Sentence modified to address reviewer's comments.
#22. Change made.
#23. Change made.
#24. The sentence has been modified to be more precise.
#25. Although the difference between productivity and efficiency is described in materials and methods, it is interesting to include it at this point to facilitate readers' understanding. Thus, the sentence has been rewritten in response to the reviewer's requests.
#26. In this case, Table 1 includes information regarding the calculation of the average number of areolas of all the explants in the trial. It is a value that indicates the average number of areolas in each type of explant depending on its initial size. To avoid any confusion to future readers, the Table 1 caption has been modified from “Average number of areolas per explant type.” to “Average number of areolas per explant type of the entire set of evaluated explants.”.
#27. Sub-section 2.1.1. renamed. In this sub-section we focused on callus productivity, considered as the production of the number of calli produced per explant. The ability to generate new shoots from callus is subsequently evaluated. We have enriched sub-section “Unexpected Morphotypes” with more discussion and relevant bibliography, but a paragraph to explain the relationship between callus production and shoot production in general terms has been included following reviewer´s request: “In any case, the relationship between callus production and shoot production may depend on various factors, such as the type of explant, the type and concentration of the PGRs used and other factors related with the in vitro culture conditions [16]. In the present work, the reduction in the number of calluses during the growth period seems to be related to an increase in the number of shoots. However, although this relationship may exist, it does not always occur in other species of cacti”. Moreover, relevant literature has been also included.
53. Astello-García, M.G.; Robles-Martínez, M.; Barba-de la Rosa, A.P.; Santos-Díaz, M. del S. Establishment of callus from Opuntia robusta Wendl., a wild and medicinal cactus, for phenolic compounds production. African J. Biotechnol. 2013, 12, 3204-3207, doi:10.4314/AJB.V12I21.
54. Cabañas-García, E.; Areche, C.; Gómez-Aguirre, Y.A.; Borquez, J.; Muñoz, R.; Cruz-Sosa, F.; Balch,
E.P.M. Biomass production and secondary metabolite identification in callus cultures of Coryphantha macromeris (Engelm.) Britton & Rose (Cactaceae), a traditional medicinal plant. South African J. Bot. 2021, 137, 1-9, doi:10.1016/J.SAJB.2020.10.002.
#28. “Apices explant” have been changed for “apical explant” along the manuscript.
#29. Student-Newman-Keuls has been performed using sample means obtained in the fifth month. The figure caption has been modified to be more precise in the procedure carried out from “Letters (a, b) above the bars represent significant differences at the fifth month for P =0.05, according to Student-Newman-Keuls.” to “Letters (a, b) above the bars represent significant differences at the fifth month based on sample means for P=0.05, according to the Student-Newman-Keuls test.”.
#30. Sentence modified to address reviewer's comments.
#31. Sub-section 2.1.2. renamed. In order to clarify the concept, the sentence “with percentages of activated areoles higher than 30%” has been modified to “with percentages of activated areoles that produced callus higher than 30%”. In this way we tried to explain that explant areoles were activated to produce callus. Besides, the initial objective of this trial was to obtain the most productive micropropagation protocol, so the productivity subsection was established before the efficiency sub-section. If the reviewer doesn't mind, we would like to continue maintaining this order so that it is consistent with the following section regarding to shoots.
#32. Change made.
#33. Table 2 caption modified as requested
#34. Additional information has been included to clarify the findings regarding to CD. Relevant literature was added to support this new paragraph.
9. Pérez-Molphe-Balch, E.; Santos-Díaz, M.D.S.; Ramírez-Malagón, R.; Ochoa-Alejo, N. Tissue culture of ornamental cacti. Sci. Agric. 2015, 72, 540-561, doi:10.1590/0103-9016-2015-0012.
16. Bouzroud, S.; El Maaiden, E.; Sobeh, M.; Devkota, K.P.; Boukcim, H.; Kouisni, L.; El Kharrassi, Y. Micropropagation of Opuntia and other cacti species through axillary shoot proliferation: A Comprehensive Review. Front. Plant Sci. 2022, 13, doi:10.3389/FPLS.2022.926653.
53. Iliev, I.; Gajdosova, A.; Libiakova, G.; Jain, S.M. Techniques of micropropagation. En Plant cell culture: essential methods; Davey, M.R., Anthony, P., Eds.; John Wiley & Sons, Ltd, Publication: United Kingdom, 2010; pp. 7-19 ISBN 0470686510.
54. Protocols for In Vitro Propagation of Ornamental Plants; Jain, S.M., Ochatt, S.J., Eds.; Methods in Molecular Biology; Humana Press, 2010; Vol. 589; ISBN 978-1-60327-390-9.
55. Mauseth, J.D. Structure function relationships in highly modified shoots of Cactaceae. Ann. Bot. 2006, 98, 901, doi:10.1093/AOB/MCL133.
#35. Figure 4 has been modified following reviewer's observations. Figure 4 caption was also improved.
#36. In order to simplify the understanding and avoid redundancies, sub-section “General Trends” was removed.
#37. Sub-section 2.2.1. renamed.
#38. Sentence replaced to the one proposed for the reviewer.
#39. Sentence replaced to the one proposed for the reviewer.
#40. Paragraph modified, separating into 3 sentences following the reviewer´s suggestions. Furthermore, scientific name of purple pitahaya has been added: Hylocereus costaricensis.
#41. Sub-section 2.2.2. renamed.
#42. Sentence modified following reviewer's advice. Reference to table included too.
#43. Sentence modified and included as part of the previous paragraph.
#44. As in the case of Table 1, Figure 5 represent sums, but letters (a, b, c, d) above the bars represent significant differences “based on sample means” at the sixth month of evaluation. This information has been added now in the figure caption to be more precise.
#45. Changes made in Figure 5.
#46. Sub-section 2.2.3. renamed.
#47. Change made.
#48. Changes have been done to refer appropriate figure/table to each sentence.
#49. Sub-section 2.3. renamed.
#50. Sentence replaced.
#51. Change made.
#52. Change made.
#53. Sentences rewritten and correctly referred to each figure/table. Initial paragraph “Bases showed a lower average production than central discs. However, BAP8_B condition produced similar shoot emergences (22) than the best ones observed in central disc results, followed by BAP4_B, which showed a total number of 16 shoots. The rest of groups showed levels below 11 shoots (Figure 5).” has been replaced by “Basal explants showed a lower average production than central discs (Supplementary Table S2). However, when evaluating specific combinations, it was observed that BAP8_B produced a shoot emergences (22) comparable to the best ones observed in central disc. The rest of groups showed levels below 16 shoots (Figure 5).”
#54. Sub-section 2.4. renamed.
#55. Sentence has been modified and simplified according to reviewer's comments. In this case, we decided to continue using “in absence of PGRs” instead “control treatment”. We mean, during the induction period, explants were subjected to different hormone concentrations with the exception of the control group. However, after this, during the growth period (from the fourth month on), all the explants were subcultured to a MS media without PGRs. That´s the reason why we used “in absence of PGRs”, to indicate that these explants were induced and, after induction, they continued their development in absence of PGRs. In contrast, control group was maintained in absence of PGRs throughout the test.
#56. The word “apex” has been replaced for “apical explants” throughout the text. The sentence “were found among the best results” has been replaced with “displayed high rates of shoot production”, as suggested.
#57. The sentences “These results conditioned the outstanding importance of the apices induced by TDZ in terms of productivity and evidenced the relevance of the existing interactions between variables. In this case, the interaction between the hormones and the type of explant (HxTE) was especially decisive, as the presence of TDZ in any of its concentrations involved a great response from the explants of the apex type. Something that was not found in the rest of the hormones and the control group.” have been re-written and simplified to facilitate reading as: “These results evidenced the relevance of the interaction between the hormones and the type of explant (HxTE). So that, the presence of TDZ (regardless of its concentration) involved the greatest responses within the overall set of the evaluated apical explants. “
#58. The sentence from lines 345-347 (older version) has been re-written: “Largest monthly increases in the number of shoots for apical explants were observed under TDZ4 and TDZ2 during the sixth month, while shoot formation under TDZ1 was greater during the fifth month (Figure 5).” Also, reference figure has been corrected. Furthermore, a space between “cactus” and the references has been included.
#59. Sub-section 2.5. renamed.
#60. The sentence from lines 423-425 (older version) has been modified: “In fact, progenies from sowings are quite heterogeneous, appearing plants with different morphologies”.
#61. Morphotype descriptions have been modified and separated into different sentences.
#62. Change made.
#63. Change made.
#64. Change made.
#65. Sub-section 2.5.1. renamed.
#66. Figure 8 caption modified following the reviewer's suggestion.
#67. Figure 9 caption modified following the reviewer's suggestion.
#68. Change made.
#69. Change made.
#70. Change made.
#71. Sub-section 2.5.2. renamed.
#72. Change made.
#73. The two suggested changes have been implemented.
#74. New references have been added to reinforce our results. Furthermore, some lines have been added (with pertinent references) related with spontaneous mutation at the end of the paragraph.
13. Giusti, P.; Vitti, D.; Fiocchetti, F.; Colla, G.; Saccardo, F.; Tucci, M. In vitro propagation of three endangered cactus species. Sci. Hortic. (Amsterdam). 2002, 95, 319-332, doi:10.1016/S0304-4238(02)00031-6.
85. Polivanova, O.B.; Bedarev, V.A. Hyperhydricity in plant tissue culture. Plants 2022, 11, doi:10.3390/PLANTS11233313.
86. Al-Mayahi, A.M.W. In vitro propagation and assessment of genetic stability in date palm as affected by chitosan and thidiazuron combinations. J. Genet. Eng. Biotechnol. 2022, 20, doi:10.1186/S43141-022-00447-9.
87. Hesami, M.; Pepe, M.; Baiton, A.; Salami, S.A.; Jones, A.M.P. New insight into ornamental applications of Cannabis: Perspectives and challenges. Plants 2022, Vol. 11, Page 2383 2022, 11, 2383, doi:10.3390/PLANTS11182383.
88. Datta, S.K. Conclusion of mutation work on ornamentals in a Nutshell. Role Mutat. Breed. Floric. Ind. 2023, 355-371, doi:10.1007/978-981-99-5675-3_14.
#75. Taking into account the length of the article and with the intention of avoiding redundancies, it was finally decided to define what could be happening with the new morphotypes in the acclimatization section. M3 did not remain stable after acclimatization and it seems obvious that it did not derive from somaclonal variation, but rather its morphology was conditioned by the in vitro culture conditions. We included “In any case, M3 is an artifact generated by the in vitro culture stress, provoked by the presence of phytohormones in the medium or due to problems related to osmotic stress.” to give an idea of the factors that could be involved in the appearance of this morphotype. However, although M4 was stable, we do not know exactly the origin of this variant and we included the sentence "It would be difficult to determine whether this variant is the result of a somaclonal variation process, due to a spontaneous mutation, or simply a rare phenotype within the existing variation that cv. Fancy presents." to highlight it.
#76. Sub-section 2.6. renamed.
#77. Paragraph has been re-written and pertinent references has been added.
91. McClelland, M.T.; Smith, M.A.L.; Carothers, Z.B. The effects of in vitro and ex vitro root initiation on subsequent microcutting root quality in three woody plants. Plant Cell. Tissue Organ Cult. 1990, 23, 115-123, doi:10.1007/BF00035831/METRICS.
92. Amghar, I.; Ibriz, M.; Ibrahimi, M.; Boudra, A.; Gaboun, F.; Meziani, R.; Iraqi, D.; Mazri, M.A.; Diria, G.; Abdelwahd, R. In vitro root induction from Argan (Argania spinosa (L.) Skeels) adventitious shoots: influence of ammonium nitrate, auxins, silver nitrate and putrescine, and evaluation of plantlet acclimatization. Plants 2021, Vol. 10, Page 1062 2021, 10, 1062, doi:10.3390/PLANTS10061062.
93. Fenning, T.; O’Donnell, M.; Preedy, K.; Bézanger, A.; Kenyon, D.; Lopez, G. The rooting ability of in vitro shoot cultures established from a UK collection of the common ash (Fraxinus excelsior L.) and their ex vitro survival. Ann. For. Sci. 2022, 79, 1-16, doi:10.1186/S13595-022-01146-8/TABLES/2.
#78. The percentages represent the number of rooted explants/total explants per treatment. So this information has been added to the text.
#79. Changes done.
#80. Changes proposed made. Besides, the auxin-like properties of the TDZ has been mentioned and a reference has been included.
95. Pai, S.R.; Desai, N.S. Effect of TDZ on various plant cultures. Thidiazuron From Urea Deriv. to Plant Growth Regul. 2018, 439-454, doi:10.1007/978-981-10-8004-3_25/TABLES/3.
#81. A new sentence regarding auxin/cytokinin ratio, with relevant literature, has been included following the reviewer's observations.
97. Su, Y.H.; Liu, Y.B.; Zhang, X.S. Auxin cytokinin interaction regulates meristem development. Mol. Plant 2011, 4, 616-625, doi:10.1093/MP/SSR007.
98. Otiende, M.A.; Fricke, K.; Nyabundi, J.O.; Ngamau, K.; Hajirezaei, M.R.; Druege, U. Involvement of the auxin-cytokinin homeostasis in adventitious root formation of rose cuttings as affected by their nodal position in the stock plant. Planta 2021, 254, doi:10.1007/S00425-021-03709-X.
#82. Sub-section 2.7. renamed.
#83. The acclimatization success rates have been expressed as percentages.
#84. Change made.
#85. On line 742, sentence has been re-written to be more specific: “Seedlings were subcultured monthly to a fresh media to promote their development.”
#86. Change made.
#87. Both changes have been implemented.
#88 and #89. A couple of sentences have been added to explain the number of explants included per petri dish, the number of replicates per treatment and total number of evaluated petri dishes; as well as how the experiment was randomized: “Six explants per type of explant and size (i.e. medium size apical, large size apical, medium size basal and large size basal explant, as well as two halves from the central disc) were cultured on each petri dish. So eight petri dishes were included per hormone treatment, which makes a total of 80 petri dishes.”
#90. As suggested by the reviewer, it has been specified for what type of variables each of the formulas have been used.
#91. Media components and pH have been specified. The typo has also been corrected.
#92. Change made.
#93. Actual value included.
#94. The last sentence has been re-written to be clearer. Furthermore, as a suggestion from the other reviewer (#2), a few sentences have been included making reference on future prospects related with our researches to close the conclusions: “So then, this first approach can be used as a guideline to continue working with variegated and colored Gymnocalycium plants, which nowadays are having an increasing relevance in the niche of the market aimed to cactus hobbyists.”

Reviewer 2 Report
Comments and Suggestions for Authors
The authors propose an article entitled “Study of the response to in vitro micropropagation of Gymnocalycium cv. Fancy (Cactaceae), a reference cultivar of commercial interest”. The manuscript is original in the conceptual idea, well structured, with new data relating on Gymnocalycium genus that have many species in the ornamental plant market and their propagation is usually carried out by traditional methods. The manuscript take into consideration the aim to establish an efficient micropropagation protocol that allows optimizing the plant obtaining processes. For this purpose, plants of two different sizes (medium and large) were used as starting material, from which three types of explants be obtained (apex, central discs and bases). A protocol based on the use of KIN at 4 µM and central discs as a starting explant was developed. Moreover, in many cases the obtained sprouts rooted successfully and their subsequent acclimatization was very effective. Furthermore, it was possible to identify unusual individuals, obtaining a new morphotype not described to date. I think the work deserves to be published. However, in the text I suggest some changes and corrections in order to have a better work.
Suggestions in bold
Abstract
Well done, no comments
1. Introduction
· Rows 30-36. The genus Gymnocalycium (Cactaceae) encompasses a plethora of cactus species that primarily extended in arid and semi-arid environments of South America, especially in Argentina, Bolivia, Brazil, Paraguay, and Uruguay [choose reference]. Their wide distribution area contributes to their ecological range and genetic diversity, reason for which Gymnocalycium genus exhibits a diverse array of botanical features and specific ecological adaptations, found in 50 and 80 taxa, in addition to a range of varieties and different forms that can be found intra-specifically [1];
· Rows 37-46. The authors declare correctly but these morphological futures are already known, please choose a reference. “Gymnocalycium species typically exhibit ………. these flowers can be solitary or produced in clusters at the apex of the stem (Figure 1c). The flower shapes, sizes, and colors vary widely among species, making them particularly appealing to pollinators [choose reference];
· Figure 1. I suggest to insert inside the pictures the letter a, b, c…….;
· Rows 53-59. Moreover, this genus garnered substantial recognition and economic importance in the ornamental plant market due to a combination of aesthetic appeal, adaptability, resilience and cultural value [choose references]. Their diverse stem shapes, striking spinations, and showy flowers make them desirable choices for both novice plant enthusiasts and experienced cactus hobbyists [choose references]. Furthermore, they require minimal care, can adapt to different climates and soil types and have the potential for long lifespans, making them interesting for a broad spectrum of gardening and landscaping projects too [choose reference];
· Please check whole introduction and add the references when necessary;
2. Results and Discussion
Few suggestions:
· Please summarize when is possible because abitually in international context the article are much shorter…;
· Figures 6 and 7. See my previous comment on Figure 1;
· Rows 536-542. The Gymnocalycium genus is included in the second appendix of the CITES agreement (Convention on International Trade in Endangered Species of Wild Fauna and Flora; http://www.cites.org), so its species are not listed as threatened or endangered, as observed in many other territories around the world [ choose 2/3 other references]. However, there are always specific populations, made up of few individuals, whose survival is threatened by numerous pressures of anthropogenic origin, such as agricultural and urban expansions, introduction of exotic grasses [choose references], use of herbicides and pesticides [57,58] in intensive agriculture [Perrino and Calabrese 2018], or their sale on the black market [59–61, Pisani et al. 2021], which lead to the result of a reduction in specific and intraspecific plant biodiversity, especially for those groups taxonomically and genetically problematics [Wagensommer et al. 2016, choose other 2/3 references]
Reference to be added:
ü Pisani, D.; Pazienza, P.; Perrino, E.V.; Caporale, D.; De Lucia, C. The Economic Valuation of Ecosystem Services of Biodiversity Components in Protected Areas: A Review for a Framework of Analysis for the Gargano National Park. Sustainability 2021, 13, 11726. Doi: 10.3390/su132111726
ü Wagensommer, R.P.; Perrino, E.V.; Albano, A.; Medagli, P.; Passalacqua, N.G. Lectotypification of four Lacaita’s names in the genus Centaurea (Asteraceae). Phytotaxa 2016, 269, 54-58. Doi: 10.11646/phytotaxa.269.1.7
3. Materials and Methods
Well done, no comments
4. Conclusion
The conclusions are good, please spend a few more words on future prospects, especially in the research field.
Author Response
Dear Reviewer,
First of all, many thanks for your contribution to improve our manuscript.
In the attached pdf file we include a cover letter addressing all the requested points from you and the first reviewer as well. So that you can have a wider perspective of the revision process.
Sincerely,
The authors

Round 2
Reviewer 1 Report
Comments and Suggestions for Authors
The authors did a great job updating the manuscript based on the suggestions. The manuscript is currently in good shape. However, there are some minor suggestions to consider before acceptance:
1. On line 89, there might be 2 spaces between "regulator" and "[16]".
2. The beginning of the sentence from line 119-122 is somewhat awkward. Please update to:
"In this experiment, medium (M) and large (L) seedlings were used as explant sources and sectioned into different explant types, including apical (A), basal (B) and central discs (CD) that were assessed. Additionally, plant growth regulators (PGRs) 6-Benzilaminopurine (BAP), Kinetin (KIN) and Thidiazuron (TDZ) were also assessed."
3. On line 579, please ensure that there is a space between "F.A.C." and "Weber" and there is only one space between "Weber" and "and".
4. For simplicity, please define "Y" variable in "Absolute Data" equation.
5. On line 876, please ensure only once space between "PGRs." and "These".
6. On line 669 and line 578, please italicize "et al.".
Comments on the Quality of English LanguageSome comments to improve grammar throughout the text are provided.
Author Response
Dear Editor,
Dear Reviewers,
Attached, please, found our revised version (2nd round) of our manuscript. Only the REVIEWER 1 suggested a few minor changes. There are no comments nor suggestions from REVIEWER 2, who recommended “accept on its current form” after our first revision. Again, we are grateful to the efforts of the reviewers and the editorial board.
We have addressed all comments and suggestions proposed by the REVIEWER 1 and, for an easy tracking, the changes have been highlighted in yellow in this revised version of the manuscript. Moreover, to facilitate the reviewer´s checking all the changes, they have been listed and explained in the next pages of this cover letter, following the same numerical order than those of the reviewer.
We hope that the new version of the manuscript meets the reviewer's expectations.
Please, do not hesitate to contact us if further information or clarification are needed
Looking forward to hearing from you
Adrián Rodríguez-Burruezo, PhD
Full Prof. Plant Genetics & Breeding
Director COMAV Institute
Universitat Politècnica de València
Response to REVIEWER 1´s comments
Comments and Suggestions for Authors:
The authors did a great job updating the manuscript based on the suggestions. The manuscript is currently in good shape. However, there are some minor suggestions to consider before acceptance:
#1. On line 89, there might be 2 spaces between "regulator" and "[16]".
Change made.
- The beginning of the sentence from line 119-122 is somewhat awkward. Please update to:
"In this experiment, medium (M) and large (L) seedlings were used as explant sources and sectioned into different explant types, including apical (A), basal (B) and central discs (CD) that were assessed. Additionally, plant growth regulators (PGRs) 6-Benzilaminopurine (BAP), Kinetin (KIN) and Thidiazuron (TDZ) were also assessed."
The sentence has been updated following reviewer´s suggestion.
- On line 579, please ensure that there is a space between "F.A.C." and "Weber" and there is only one space between "Weber" and "and".
The spaces between words have been corrected.
- For simplicity, please define "Y" variable in "Absolute Data" equation.
“Y” has been defined as each “Single data” to facilitate understanding
- On line 876, please ensure only once space between "PGRs." and "These".
Spaces between words has been corrected.
- On line 669 and line 578, please italicize "et al.".
Both “et al.” Have been changed to italics.
Comments on the Quality of English Language
Some comments to improve grammar throughout the text are provided.
We did not found any attached file from the reviewer in this sense. Nevertheless, the whole document has been revised by one colleague of us. Lecturer of the Department of Languages (English) at the UPV. She found the manuscript fine, with the only exception of very few minor mistakes, which have been revised in this second version.
Reviewer 2 Report
Comments and Suggestions for Authors
Dear authors, I appreciate the hard work done following my suggestions. In my opinion the manuscript is now able to be published.
Regards,
reviewer
Author Response
The author found the manuscript OK after our first revision. Therefore, no new changes have been done in this regard